# Reinforcement Learning with Sparse Rewards using Guidance from Offline Demonstration

**Desik Rengarajan, Gargi Vaidya, Akshay Sarvesh, Dileep Kalathil & Srinivas Shakkottai**
Department of Electrical and Computer Engineering, Texas A&M University
{desik,gargivaidya,sarvesh,dileep.kalathil,sshakkot}@tamu.edu

## Abstract

A major challenge in real-world reinforcement learning (RL) is the sparsity of reward feedback. Often, what is available is an intuitive but sparse reward function that only indicates whether the task is completed partially or fully. However, the lack of carefully designed, fine grain feedback implies that most existing RL algorithms fail to learn an acceptable policy in a reasonable time frame. This is because of the large number of exploration actions that the policy has to perform before it gets any useful feedback that it can learn from. In this work, we address this challenging problem by developing an algorithm that exploits the offline demonstration data generated by a sub-optimal behavior policy for faster and efficient online RL in such sparse reward settings. The proposed algorithm, which we call the Learning Online with Guidance Offline (LOGO) algorithm, merges a policy improvement step with an additional policy guidance step by using the offline demonstration data. The key idea is that by obtaining guidance from - not imitating - the offline data, LOGO orients its policy in the manner of the sub-optimal policy, while yet being able to learn beyond and approach optimality. We provide a theoretical analysis of our algorithm, and provide a lower bound on the performance improvement in each learning episode. We also extend our algorithm to the even more challenging incomplete observation setting, where the demonstration data contains only a censored version of the true state observation. We demonstrate the superior performance of our algorithm over state-of-the-art approaches on a number of benchmark environments with sparse rewards and censored state. Further, we demonstrate the value of our approach via implementing LOGO on a mobile robot for trajectory tracking and obstacle avoidance, where it shows excellent performance.

## 1 Introduction

Reinforcement Learning (RL) is being considered for application to a variety of real-world settings that have very large state-action spaces, but even under the availability of accurate simulators determining how best to explore the environment is a major challenge. Oftentimes, the problem setting is such that intuitively obvious reward functions are quite sparse. Such rewards typically correspond to the attainment of a particular task, such as a robot attaining designated way points, with little fine-grain reward feedback on the intermediate steps taken to complete these tasks. The sparsity of rewards implies that existing RL approaches often do not make much headway into learning viable policies unless provided with carefully hand-tuned reward feedback by a domain expert.

Many of these problems correspond to systems where data has been gathered over time using an empirically determined (sub-optimal) behavior policy, which contains information important to help bootstrap learning. An additional caveat is that this behavior data might only contain measurements of a subset of the true state. Is it possible to design an algorithm that has principled policy optimization for efficient learning, while also being able to explore by utilizing the data derived from a behavior policy? A natural candidate to begin with is the framework of policy gradient algorithms (Schulman et al., 2015; Lillicrap et al., 2016; Haarnoja et al., 2018), which performs really well in the dense reward setting. Can such a policy gradient algorithm be aligned with a behavior policy for guided exploration that is critical in the sparse reward setting?

In this work, we propose a principled approach for using the well known trust region policy optimization (TRPO) algorithm (Schulman et al., 2015) with offline demonstration data for guidance in a sparse reward setting. Our choice of the TRPO approach is motivated by its analytical tractability and superior performance. Our key insight is that we can utilize the trust region approach while being guided by the behavior data by virtue of two steps. The first step is identical to traditional TRPO to generate a candidate policy. In the second step, the objective is to find a policy closest to the behavior policy, subject to it being in the trust region of the candidate generated in the first step. Thus, the second step ensures that the policy chosen is always guided by the behavior policy, but the level of alignment with the behavior policy can be reduced by shrinking the trust region over time. We call our approach Learning Online with Guidance Offline (LOGO) to capture this two-step procedure. This principled approach enables us to provide analytical performance guarantees while achieving superior performance in a number of benchmark problems.

Our main results are as follows: (i) LOGO can guarantee a performance improvement as long as the behavior policy is able to offer an advantage. Reducing the alignment with the behavior policy over time allows us to extract this advantage and then smoothly move beyond to find near-optimal policies, (ii) we provide a generalized version of the Performance Difference Lemma (Kakade & Langford, 2002) wherein the stage reward function can depend on the policy itself, and use this result to determine a surrogate objective function for step 2 of LOGO that is straightforward to evaluate. This allows us to implement LOGO in the manner of two TRPO-like steps, enabling us to leverage the TRPO code base, (iii) we show on standard MuJoCo environments that LOGO trained with sparse rewards can attain nearly the same performance as an optimal algorithm trained with dense rewards. We go further and show that this excellent performance carries over to waypoint tracking by a robot over Gazebo simulations and real-world TurtleBot experiments, (iv) finally, we show that LOGO can also be used in the case where the demonstration data only contains a censored version of the true state (incomplete state information) by simply adding a projection to the available subset of the state space in step 2 of the algorithm. Again, we provide supporting evidence via excellence in both MuJoCo simulations, as well as obstacle avoidance by a TurtleBot that is trained with a behavior policy without a Lidar (censored state), but is tested with it (full state).

## 1.1 RELATED WORK

Our work is mainly related to two RL research areas:

**Imitation Learning (IL):** The goal of an IL algorithm is to imitate an (expert) policy using the demonstration data generated by that policy. Behavior cloning (BC) is a simple IL approach where the expert policy is estimated from the demonstration data using supervised learning. BC algorithms, however, suffer from the problem of distribution shift (Ross et al., 2011). Inverse reinforcement learning (IRL) algorithms (Ng & Russell, 2000; Ziebart et al., 2008) estimate a reward function from the demonstration data and solve a forward RL problem using this reward function. Generative adversarial imitation learning (GAIL) (Ho & Ermon, 2016) avoids the reward estimation problem by formulating the IL problem as a distribution matching problem, and provides an implicit reward for the RL algorithm using a discriminator (Goodfellow et al., 2014). Most IL algorithms do not use reward feedback from the environment, and hence are restricted to the performance of the policy that generated the demonstration data. Our approach is different from pure IL, and we leverage online RL with (sparse) reward feedback for efficient learning.

**Learning from Demonstration (LfD):** The key idea of the LfD algorithms is to use demonstration data to aid online learning (Schaal et al., 1997). Many works propose to exploit demonstration data by adding it to the replay buffer with a prioritized replay mechanism to accelerate learning (Hester et al., 2018; Vecerik et al., 2017; Nair et al., 2018). Rajeswaran et al. (2018) combine a policy gradient algorithm with demonstration data by using a mix of behavior cloning and online RL fine tuning. Nair et al. (2020) propose AWAC algorithm to accelerate online RL by leveraging large amounts of offline data with associated rewards. Different from this, LOGO does not need the reward observations. Moreover, we give provable guarantees on the performance of LOGO whereas AWAC does not have any such provable guarantee (further details are provided in Appendix G). Kim et al. (2013) propose a framework to integrate LfD and approximate policy iteration by formulating a coupled constraint convex optimization problem, where the expert demonstrations define a set of linear constraints. This approach is, however, limited to small problems with a discrete action space. The closest to our work is the PofD algorithm proposed by Kang et al. (2018). PofD modifies the reward function by taking a weighted combination of the sparse reward from the online interaction and an implicit

reward obtained from the demonstration data using a discriminator. Very different from this, we propose an intuitive and principled approach of using the offline demonstration data for guiding the online exploration during the initial phase of learning. Our two step approach enables us to provide a rigorous performance guarantee for the proposed LOGO algorithm, and to leverage trust region-based approaches to solve high dimensional problems with even sparser settings. We provide an extensive comparison between our algorithm and PofD in Section 5.

**Offline RL:** Recently, there has been many interesting works in the area of offline RL (Kumar et al., 2019; Fujimoto et al., 2019; Siegel et al., 2020; Wu et al., 2019a) which focus on learning a policy using *only* the offline data *without* any online learning or online fine-tuning. Different from these, LOGO is an *online* RL algorithm (further details are provided in Appendix G).

## 2 PRELIMINARIES AND PROBLEM SETTING

### 2.1 PRELIMINARIES

A Markov Decision Process (MDP) is denoted as a tuple $< \mathcal{S}, \mathcal{A}, R, P, \gamma >$, where $\mathcal{S}$ is the state space, $\mathcal{A}$ is the action space, $R : \mathcal{S} \times \mathcal{A} \to \mathbb{R}$ is the reward function, $P : \mathcal{S} \times \mathcal{A} \times \mathcal{S} \to [0, 1]$ is the transition probability function with $P(s'|s, a)$ giving the probability of transitioning to state $s'$ when action $a$ is taken at state $s$, and $\gamma \in (0, 1)$ is the discount factor. A policy $\pi$ is a mapping from $\mathcal{S}$ to probability distribution over $\mathcal{A}$, with $\pi(s, a)$ specifying the probability of taking action $a$ in state $s$. A policy $\pi$ can generate state-action trajectory $\tau$, where $\tau = (s_0, a_0, s_1, a_1, \ldots), s_0 \sim \mu, a_t \sim \pi(s_t, \cdot), s_{t+1} \sim P(\cdot|s_t, a_t)$ and $\mu$ is the initial state distribution. The infinite horizon discounted return of policy $\pi$ is defined as $J_R(\pi) = \mathbb{E}_{\tau \sim \pi} \left[ \sum_{t=0}^{\infty} \gamma^t R(s_t, a_t) \right]$. The goal of a reinforcement learning algorithm is to learn the optimal policy $\pi^\star = \arg\max_\pi J_R(\pi)$.

The value function of a policy $\pi$ defined as $V_R^\pi(s) = \mathbb{E}_{\tau \sim \pi} \left[ \sum_{t=0}^{\infty} \gamma^t R(s_t, a_t)|s_0 = s \right]$, is the expected cumulative discounted reward obtained by following the policy $\pi$ starting from the state $s$. The action-value function of a policy $\pi$ is defined similarly as $Q_R^\pi(s, a) = \mathbb{E}_{\tau \sim \pi} \left[ \sum_{t=0}^{\infty} \gamma^t R(s_t, a_t)|s_0 = s, a_0 = a \right]$. The advantage function is defined as $A_R^\pi(s, a) = Q_R^\pi(s, a) - V_R^\pi(s)$. The discounted state visitation distribution for the policy $\pi$, denoted as $d^\pi$, is defined as $d^\pi(s) = (1 - \gamma) \sum_{t=0}^{\infty} \gamma^t \mathbb{P}(s_t = s|\pi)$, where the probability is defined with respect to the randomness induced by $\pi, P$ and $\mu$.

**Definitions and Notations:** The Kullback-Leibler (KL) divergence between two distribution $p$ and $q$ is defined as $D_{\mathrm{KL}}(p, q) = \sum_x p(x) \log \frac{p(x)}{q(x)}$. The average KL divergence between two policies $\pi_1$ and $\pi_2$ with respect to $d^{\pi_1}$ is defined as $D_{\mathrm{KL}}^{\pi_1}(\pi_1, \pi_2) = \mathbb{E}_{s \sim d^{\pi_1}} [D_{\mathrm{KL}}(\pi_1(s, \cdot), \pi_2(s, \cdot))]$. The maximum KL divergence between two polices $\pi_1$ and $\pi_2$ is defined as $D_{\mathrm{KL}}^{\max}(\pi_1, \pi_2) = \max_s D_{\mathrm{KL}}(\pi_1(s, \cdot), \pi_2(s, \cdot))$. The total variation (TV) distance between two distribution $p$ and $q$ is defined as $D_{\mathrm{TV}}(p, q) = (1/2) \sum_x |p(x) - q(x)|$. The average TV distance and the maximum TV distance between two polices $\pi_1$ and $\pi_2$ are defined as $D_{\mathrm{TV}}^{\pi_1}(\pi_1, \pi_2) = \mathbb{E}_{s \sim d^{\pi_1}} [D_{\mathrm{TV}}(\pi_1(s, \cdot), \pi_2(s, \cdot))]$ and $D_{\mathrm{TV}}^{\max}(\pi_1, \pi_2) = \max_s D_{\mathrm{TV}}(\pi_1(s, \cdot), \pi_2(s, \cdot))$, respectively.

### 2.2 PROBLEM SETTING

As described in the introduction, our goal is to develop an algorithm that can exploit offline demonstration data generated using a sub-optimal behavior policy for faster and efficient online reinforcement learning in a sparse reward setting. Formally, we assume that the algorithm has access to the demonstration data generated by a sub-optimal behavior policy $\pi_{\mathrm{b}}$. We first consider the setting in which the demonstration data has complete state observation. In this setting, the demonstration data has the form $\mathcal{D} = \{\tau^i\}_{i=1}^n$, where $\tau^i = (s_1^i, a_1^i, \ldots, s_T^i, a_T^i), \tau_i \sim \pi_{\mathrm{b}}$. Later, we also propose an extension to the incomplete observation setting in which only a censored version of the true state is available in demonstration data. More precisely, instead of the complete state observation $s_t^i$, the demonstration data will only contain $\tilde{s}_t^i = o(s_t^i)$, where $o(\cdot)$ is a projection to a lower dimensional subspace. We represent the incomplete demonstration data as $\tilde{\mathcal{D}} = \{\tilde{\tau}^i\}_{i=1}^n$, where $\tilde{\tau}^i = (\tilde{s}_1^i, a_1^i, \ldots, \tilde{s}_T^i, a_T^i)$.

We make the following assumption about the behavior policy $\pi_{\mathrm{b}}$.

**Assumption 1.** *In the initial episodes of learning, $\mathbb{E}_{a \sim \pi_{\mathrm{b}}} [A_R^{\pi_k}(s, a)] \geq \beta > 0, \forall s$, where $\pi_k$ is the learning policy employed by the algorithm in the $k$th episode of learning.*

Intuitively, the above assumption implies that taking action according to $\pi_b$ will provide a higher advantage than taking actions according to $\pi_k$ because $\mathbb{E}_{a \sim \pi_k}[A_R^{\pi_k}(s, a)] = 0$. This is a reasonable assumption, since the behavior policies currently in use in many systems are likely to perform much better than an untrained policy. We also note that a similar assumption is made by Kang et al. (2018) to formalize the notion of a useful behavior policy. We emphasize that $\pi_b$ need not be the optimal policy, and $\pi_b$ could be such that $J_R(\pi_b) < J_R(\pi^\star)$. Our proposed algorithm learns a policy that performs better than $\pi_b$ through online learning with guided exploration.

## 3 ALGORITHM AND PERFORMANCE GUARANTEE

### 3.1 ALGORITHM

In this section, we describe our proposed LOGO algorithm. Each iteration of our algorithm consists of two steps, namely a policy improvement step and a policy guidance step. In the following, $\pi_k$ denotes the policy after $k$ iterations.

**Step 1: Policy Improvement:** In this step, the LOGO algorithm performs a one step policy improvement using the Trust Region Policy Optimization (TRPO) approach (Schulman et al., 2015). This can be expressed as

$$\pi_{k+1/2} = \arg\max_\pi \ \mathbb{E}_{s \sim d^{\pi_k}, a \sim \pi}[A_R^{\pi_k}(s, a)] \quad \text{s.t.} \quad D_{\mathrm{KL}}^{\pi_k}(\pi, \pi_k) \leq \delta. \tag{1}$$

The TRPO update finds the policy $\pi_{k+1/2}$ that maximizes the objective while constraining this maximizing policy to be within the *trust region* around $\pi_k$ defined as $\{\pi : D_{\mathrm{KL}}^{\pi_k}(\pi, \pi_k) \leq \delta\}$. The TRPO approach provides a provable guarantee on the performance improvement, as stated in Proposition 1. We omit the details as this is a standard approach in the literature.

**Step 2: Policy Guidance:** While the TRPO approach provides an efficient online learning strategy in the dense reward setting, it fails when the reward structure is sparse. In particular, it fails to achieve any significant performance improvement for a very large number of episodes in the initial phase of learning due to the lack of useful reward feedback that is necessary for efficient exploration. We propose to overcome this challenge by providing policy guidance using offline demonstration data after each step of the TRPO update. The policy guidance step is given as

$$\pi_{k+1} = \arg\min_\pi \ D_{\mathrm{KL}}^\pi(\pi, \pi_b) \quad \text{s.t.} \quad D_{\mathrm{KL}}^{\max}(\pi, \pi_{k+1/2}) \leq \delta_k. \tag{2}$$

Intuitively, the policy guidance step aids learning by selecting the policy $\pi_{k+1}$ in the direction of the behavior policy $\pi_b$. This is achieved by finding a policy that minimizes the KL divergence w.r.t. $\pi_b$, but at the same time lies inside the trust region around $\pi_{k+1/2}$ defined as $\{\pi : D_{\mathrm{KL}}^{\max}(\pi, \pi_{k+1/2}) \leq \delta_k\}$. This trust region-based policy guidance is the key idea that distinguishes LOGO from other approaches. In particular, this approach gives LOGO two unique advantages over other state-of-the-art algorithms.

First, unlike imitation learning that tries to mimic the behavior policy by directly minimizing the distance (typically KL/JS divergence) between it and the current policy, LOGO only uses the behavior policy to guide initial exploration. This is achieved by starting the guidance step with a large value of the trust region $\delta_k$, and gradually decaying it according to an adaptive update rule (specified in Appendix F) as learning progresses. This novel approach of trust region based policy improvement and policy guidance enables LOGO to both learn faster in the initial phase by exploiting demonstration data, and to converge to a better policy than the sub-optimal behavior policy.

Second, from an implementation perspective, the trust region based approach allows to approximate the objective by a surrogate function that is amenable to sample based learning (see Proposition 2). This allows us to implement LOGO in the manner of two TRPO-like steps, enabling us to leverage the TRPO code base. Details are given Section 4.

### 3.2 PERFORMANCE GUARANTEE

We derive a lower bound on the performance improvement, $J_R(\pi_{k+1}) - J_R(\pi_k)$ for LOGO in each learning episode. We analyze the policy improvement step and policy guidance step separately. We begin with the performance improvement due to the TRPO update (policy improvement step). The following result and its analysis are standard in the literature and we omit the details and the proof.

**Proposition 1** (Proposition 1, (Achiam et al., 2017)). *Let $\pi_k$ and $\pi_{k+1/2}$ are related by (1). Then,*

$$J_R(\pi_{k+1/2}) - J_R(\pi_k) \geq -\sqrt{2\delta}\gamma\epsilon_{R,k}/(1-\gamma)^2, \tag{3}$$

*where $\epsilon_{R,k} = \max_{s,a}|A_R^{\pi_k}(s,a)|$.*

We now give the following result which can be used to obtain a lower bound on the performance improvement due to the policy guidance step.

**Lemma 1.** *Let $\pi_{k+1/2}$ be a policy that satisfies Assumption 1. Then, for any policy $\pi$,*

$$J_R(\pi) - J_R(\pi_{k+1/2}) \geq (1-\gamma)^{-1}\beta - (1-\gamma)^{-1}\epsilon_{R,k+1/2}\sqrt{2D_{\mathrm{KL}}^\pi(\pi, \pi_{\mathrm{b}})}, \tag{4}$$

*Where $\epsilon_{R,k+1/2} = \max_{s,a}|A_R^{\pi_{k+1/2}}(s,a)|$.*

*Remark* 1. Lemma 1 also gives an intuitive explanation for our proposed policy guidance step. The policies used in the initial phase of the learning can be far from the optimal. So, it is reasonable to assume that $\pi_k$ (and hence $\pi_{k+1/2}$) satisfies Assumption 1 in the initial phase of learning. Then, minimizing $D_{\mathrm{KL}}^\pi(\pi, \pi_{\mathrm{b}})$ can get a non-negative lower bound in (4), which will imply that the performance of $\pi_{k+1}$ is better than $\pi_{k+1/2}$. This is indeed the idea behind the policy guidance step.

Combining the results of Proposition 1 and Lemma 1, and with some more analysis, we get the following performance improvement guarantee for the LOGO algorithm.

**Theorem 1.** *Let $\pi_k$ and $\pi_{k+1/2}$ are related by (1) and let $\pi_{k+1/2}$ and $\pi_{k+1}$ are related by 2. Let $\epsilon_{R,k}$ and $\epsilon_{R,k+1/2}$ be as defined in Proposition 1 and Lemma 1, respectively. Let $R_{\max} = \max_{s,a}|R(s,a)|$.*
*(i) If $\pi_{k+1/2}$ satisfies Assumption 1, then*

$$J_R(\pi_{k+1}) - J_R(\pi_k) \geq \frac{-\sqrt{2\delta}\gamma\epsilon_{R,k}}{(1-\gamma)^2} + \frac{\beta}{(1-\gamma)} - \frac{\epsilon_{R,k+1/2}}{(1-\gamma)}\sqrt{2D_{\mathrm{KL}}^\pi(\pi_{k+1}, \pi_{\mathrm{b}})}. \tag{5}$$

*(ii) If $\pi_{k+1/2}$ does not satisfy Assumption 1, then*

$$J_R(\pi_{k+1}) - J_R(\pi_k) \geq -(\sqrt{2\delta}\gamma\epsilon_{R,k} + 3R_{\max}\delta_k)/(1-\gamma)^2. \tag{6}$$

*Remark* 2. In the initial phase of learning when the baseline policy is better than the current policy, the policy guidance step can add a non-negative term to the standard lower bound obtained by the TRPO approach, as shown in (5). This indicates faster learning in the initial phase as compared to the naive TRPO approach. The policy improvement guarantee during the later phase of learning is given by (6), which shows that LOGO achieves similar performance guarantee as TRPO when $\delta_k = O(\sqrt{\delta})$. We ensure this by decreasing the value of $\delta_k$ as the learning progresses. Thus, Theorem 1 clearly shows the key advantage of the LOGO algorithm achieved by the novel combination of the policy improvement step and the policy guidance step.

## 4 PRACTICAL ALGORITHM

We first develop an approximation to the policy guidance step (2) that is amenable to sample-based learning and can scale to policies paramaterized by neural networks. This step involves minimizing $D_{\mathrm{KL}}^\pi(\pi, \pi_{\mathrm{b}})$ under a trust region constraint. However, this is not easy to solve directly by a sample-based learning approach, because estimating it requires samples generated according to any possible $\pi$, which is clearly infeasible. To overcome this issue, inspired by the surrogate function idea used in the TRPO algoirthm (Schulman et al., 2015), we derive a surrogate function for $D_{\mathrm{KL}}^\pi(\pi, \pi_{\mathrm{b}})$ that can be estimated using only the samples from the policy $\pi_{k+1/2}$.

For deriving a surrogate function for $D_{\mathrm{KL}}^\pi(\pi, \pi_{\mathrm{b}})$ that is amenable to sample-based learning, we first define the policy dependent reward function $C_\pi$ as $C_\pi(s,a) = \log(\pi(s,a)/\pi_{\mathrm{b}}(s,a))$. Using $C_\pi$, we can also define the quantities $J_{C_\pi}(\tilde{\pi}), V_{C_\pi}^{\tilde{\pi}}, Q_{C_\pi}^{\tilde{\pi}}$ and $A_{C_\pi}^{\tilde{\pi}}$ for any policy $\tilde{\pi}$, exactly as defined in Section 2.1 by replacing $R$ by $C_\pi$. Using these notations, we now present an interesting result which we call the performance difference lemma for policy dependent reward function.

**Lemma 2.** *For any policies $\pi$ and $\tilde{\pi}$,*

$$J_{C_\pi}(\pi) - J_{C_{\tilde{\pi}}}(\tilde{\pi}) = (1-\gamma)^{-1}\mathbb{E}_{s\sim d^\pi, a\sim\pi(s,\cdot)}[A_{C_{\tilde{\pi}}}^{\tilde{\pi}}(s,a)] + (1-\gamma)^{-1}D_{\mathrm{KL}}^\pi(\pi, \tilde{\pi}). \tag{7}$$

Note that the above result has an additional term, $(1 - \gamma)^{-1} D_{\mathrm{KL}}^\pi(\pi, \tilde{\pi})$, compared to the standard performance difference lemma (Kakade & Langford, 2002). In addition to being useful for analyzing our algorithm, we believe that the above result may also be of independent interest.

We now make an interesting observation that $J_{C_\pi}(\pi) = (1 - \gamma)^{-1} D_{\mathrm{KL}}^\pi(\pi, \pi_\mathrm{b})$ (proof is given in the Appendix), which can be used with Lemma 2 to derive the surrogate function given below.

**Proposition 2.** *Let $\pi_{k+1/2}$ be as given in* (1). *Then, for any policy $\pi$ that lies in the trust region around $\pi_{k+1/2}$ defined as $\{\pi : D_{\mathrm{KL}}^{\max}(\pi, \pi_{k+1/2}) \leq \delta_k\}$, we have*

$$D_{\mathrm{KL}}^\pi(\pi, \pi_\mathrm{b}) \leq \alpha_k + \mathbb{E}_{s \sim d^{\pi_{k+1/2}}, a \sim \pi(s, \cdot)}[A_{C_{\pi_{k+1/2}}}^{\pi_{k+1/2}}(s, a)] + \gamma(1 - \gamma)^{-1} \epsilon_{\pi,k} \sqrt{2\delta_k} + \delta_k, \quad (8)$$

*where $\alpha_k = D_{\mathrm{KL}}^{\pi_{k+1/2}}(\pi_{k+1/2}, \pi_\mathrm{b})$, $\epsilon_{\pi,k} = \max_{s,a} |A_{C_{\pi_{k+1/2}}}^{\pi_{k+1/2}}(s, a)|$.*

Now, instead of minimizing $D_{\mathrm{KL}}^\pi(\pi, \pi_\mathrm{b})$, we will minimize the upper bound as a surrogate objective function. Note that only one term in this surrogate objective function depends on $\pi$, and that term can be estimated using only samples from $\pi_{k+1/2}$. Thus, we will approximate the policy guidance step as

$$\pi_{k+1} = \arg\min_\pi \ \mathbb{E}_{s \sim d^{\pi_{k+1/2}}, a \sim \pi(s, \cdot)}[A_{C_{\pi_{k+1/2}}}^{\pi_{k+1/2}}(s, a)] \quad \text{s.t.} \quad D_{\mathrm{KL}}^{\max}(\pi, \pi_{k+1/2}) \leq \delta_k. \quad (9)$$

Since $D_{\mathrm{KL}}^{\max}$ is difficult to implement in practice, we will further approximate it by average KL divergence $D_{\mathrm{KL}}^{\pi_{k+1/2}}(\pi, \pi_{k+1/2})$. We note that this is a standard approach used in the TRPO algorithm.

We can now put steps (1) and (9) together to obtain the full algorithm. However, as the policies are represented by large neural networks, solving them exactly is challenging. Hence, for sufficiently small $\delta, \delta_k$, we can further approximate the objective functions and constraints by a Taylor series expansion to obtain readily implementable update equations. This is a standard approach (Schulman et al., 2015; Achiam et al., 2017), and we only present the final result below.

Consider the class of policies $\{\pi_\theta : \theta \in \Theta\}$ where $\theta$ is the parameter of the policy. Let $\theta_k$ be the parameter corresponding to policy $\pi_k$. Then, the Taylor series expansion-based approximate solution of (1) and (9) yields the final form of LOGO as follows:

$$\theta_{k+1/2} = \theta_k + \sqrt{\frac{2\delta}{g_k^T F_k^{-1} g_k}} F_k^{-1} g_k, \quad \theta_{k+1} = \theta_{k+1/2} - \sqrt{\frac{2\delta_k}{h_k^T L_k^{-1} h_k}} L_k^{-1} h_k, \quad (10)$$

where $g_k = \nabla_\theta \ \mathbb{E}_{s \sim d^{\pi_k}, a \sim \pi_\theta(s, \cdot)}[A_R^{\pi_k}(s, a)]$, $F_k = \nabla_\theta^2 \ D_{\mathrm{KL}}^{\pi_k}(\pi_\theta, \pi_k)$, and $h_k = \nabla_\theta \ \mathbb{E}_{s \sim d^{\pi_{k+1/2}}, a \sim \pi_\theta(s, \cdot)}[A_{C_{\pi_{k+1/2}}}^{\pi_{k+1/2}}(s, a)]$, $L_k = \nabla_\theta^2 \ D_{\mathrm{KL}}^{\pi_{k+1/2}}(\pi_\theta, \pi_{k+1/2})$.

While it is straightforward to compute $A_{C_\pi}^\pi$ for any policy when the form of the baseline policy $\pi_\mathrm{b}$ is known, it is more challenging when only the demonstration data $\mathcal{D}$ generated according to $\pi_\mathrm{b}$ is available. We overcome this challenge by training a discriminator using the demonstration data $\mathcal{D}$ and the data $\mathcal{D}_{k+1/2}$ generated by the policy $\pi_{k+1/2}$ (Goodfellow et al., 2014; Ho & Ermon, 2016; Kang et al., 2018) that will approximate the policy dependent reward function $C_{\pi_{k+1/2}}$. Further details on training this discriminator and a concise form of the algorithm are given in Appendix D.

## 4.1 EXTENSION TO INCOMPLETE OBSERVATION SETTING

We now discuss how to extend LOGO to the setting where the behavior policy data contains only incomplete state observations. For instance, consider the problem of learning a policy for a mobile robot to reach a target point without colliding with any obstacles. Collision avoidance requires sensing the presence of obstacles, which is typically achieved by camera/Lidar sensors. Learning an RL policy for this problem requires a high fidelity simulator that models various sensors and their interaction with the dynamics. Such high fidelity simulators are, however, typically slow and difficult to parallelize, and training an RL policy using such a simulator in a sparse reward environment can be very time consuming and computationally expensive. Often, it is much easier to train an RL policy for the trajectory tracking problem using only a simple kinematics simulator model which only has a lower dimensional state space compared to the original problem. In particular, such a kinematics simulator will only have the position and velocity of the robot as the state instead of the true state with

high dimensional camera/Lidar image. Can we use the demonstration data or behavior policy from this low dimensional simulator to accelerate the RL training in a high dimensional/fidelity simulator in sparse reward environments? We answer this question affirmatively using a simple idea to extend LOGO into such incomplete observation settings.

Since the behavior policy appears in LOGO only through a policy dependent reward function $C_\pi(s, a) = \log(\pi(s, a)/\pi_{\mathrm{b}}(s, a))$, we propose to replace this with a form that can handle the incomplete observation setting. For any state $s \in \mathcal{S}$, let $\tilde{s} = o(s)$ be its projection to a lower dimensional state space $\tilde{\mathcal{S}}$. As explained in the mobile robot example above, a behavior policy $\tilde{\pi}_{\mathrm{b}}$ in the incomplete observation setting can be interpreted as mapping from $\tilde{\mathcal{S}}$ to the set of probability distributions over $\mathcal{A}$. We can then replace $C_\pi(s, a)$ in the LOGO algorithm with $\tilde{C}_\pi(s, a)$ defined as $\tilde{C}_\pi(s, a) = \log(\pi(s, a)/\tilde{\pi}_{\mathrm{b}}(o(s), a))$. When only the demonstration data with incomplete observation is available instead of the representation of the behavior policy, we propose to train a discriminator to estimate $\tilde{C}_\pi$. Let $\tilde{\mathcal{D}} = \{\tilde{\tau}^i\}_{i=1}^n$ be the demonstration with incomplete observation, where $\tilde{\tau}^i = (\tilde{s}_1^i, a_1^i, \ldots, \tilde{s}_T^i, a_T^i)$. Then we train a discriminator using $\tilde{\mathcal{D}}$ and $\mathcal{D}_\pi$ to estimate $\tilde{C}_\pi$.

## 5 EXPERIMENTS

We now evaluate LOGO from two perspectives: (i) Can LOGO learn near-optimally in a sparse reward environment when guided by demonstration data generated by a sub-optimal policy? (ii) Can LOGO retain near-optimal performance when guided by sub-optimal and incomplete demonstration data with sparse rewards? We perform an exhaustive performance analysis of LOGO, first through simulations under four standard (sparsified) environments on the widely used MuJoCo platform (Todorov et al., 2012). Next, we conduct simulations on the Gazebo simulator (Koenig & Howard, 2004) using LOGO for way-point tracking by a robot in environments with and without obstacles, with the only reward being attainment of way points. Finally, we transfer the trained models to a real-world TurtleBot robot (Amsters & Slaets, 2019) to demonstrate LOGO in a realistic setting.

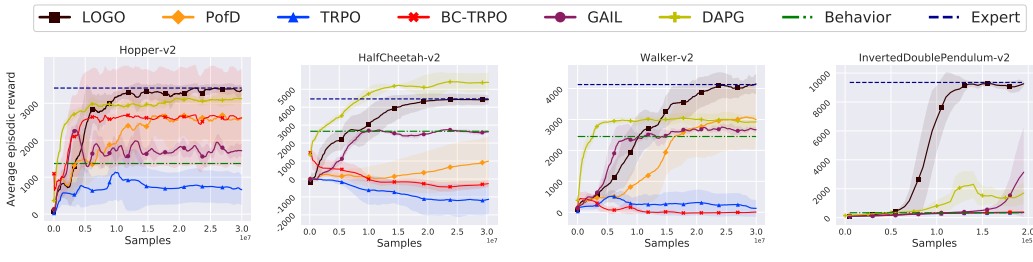

(a) Evaluation on MuJoCo with full offline observation.

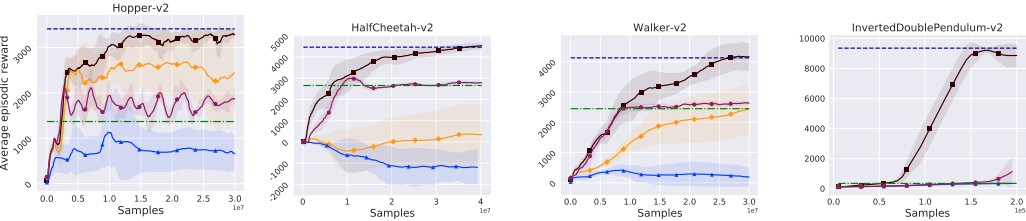

(b) Evaluation on MuJoCo with incomplete offline observation.

Figure 1: Evaluation of algorithms on four sparse reward MuJoCo environments with complete (**1a**) and incomplete (**1b**) offline data. The solid line corresponds to the mean over five trials with different seeds and the shaded region corresponds to the standard deviation over the trials.

In what follows, we present a summary of our experiments[1] and results. We remark that LOGO is relatively easy to implement and train, since its two TRPO-like steps imply that we can utilize much of a TRPO code base.

---

[1]code base and a video of the TurtleBot experiments: https://github.com/DesikRengarajan/LOGO

## 5.1 MuJoCo Simulations

In our first set of experiments, we consider four standard environments using the MuJoCo platform. We introduce sparsity by reducing the events at which reward feedback is provided. Specifically, for Hopper, HalfCheetah and Walker2d, we provide a reward of $+1$ at each time only after the agent moves forward over $2, 20$, and $2$ units from its initial position, respectively. For InvertedDoublePendulum, we introduce sparsity by providing a reward only at the end of the episode.

Besides LOGO, our candidate algorithms are as follows: (i) **Expert:** We train TRPO in the dense reward environment to provide the optimal baseline, (ii) **Behavior:** We use a partially trained expert that is still at a sub-optimal stage of learning to provide behavior data, (iv) **GAIL:** We use Generative Adversarial Imitation Learning (Ho & Ermon, 2016), which attempts to imitate the Behavior policy (iii) **TRPO:** We directly use TRPO in the sparse reward setting without any guidance from behavior data (iv) **POfD:** We use Policy Optimization from Demonstration (Kang et al., 2018) as a heuristic approach to exploiting behavior data. (v) **BC-TRPO:** We warm start TRPO by performing behavior cloning (BC) on the sub-optimal behavior data. (vi) **DAPG:** We use Demo Augmented Policy Gradient (Rajeswaran et al., 2018) which warm starts Natural Policy Gradient (NPG) algorithm using BC, and fine tunes it online using behavior data in a heuristic manner. Note that for all algorithms, we evaluate the final performance in the corresponding dense reward environment provided by OpenAI Gym, which provides a standardized way of comparing their relative merits.

**Setting 1: Sparse Rewards.** We compare the performance of our candidate algorithms in the sparse reward setting in Figure 1a, which illustrates their rewards during training. As expected, TRPO fails to make much meaningful progress during training, while GAIL can at best attain the same performance as the sub-optimal behavior policy. While BC-TRPO benefits from warm starting, it fails to learn later due to the absence of online guidance as in the case of LOGO. POfD and LOGO both use the behavior data to boot strap learning. POfD suffers from the fact that it is influenced throughout the learning process by the behavior data, which prevents it from learning the best policy. However, LOGO's nuanced exploration using the demonstration data only as guidance enables it to quickly attain optimality. LOGO outperforms DAPG in all but one environment.

**Setting 2: Sparse Rewards and Incomplete State.** We next consider sparse rewards along with reducing the revealed state dimensions in the behavior data of Setting 1. These state dimensions are selected by eliminating state dimensions revealed until the expert TRPO with dense rewards starts seeing reduced performance. This ensures that we are pruning valuable information. Since GAIL, POfD and LOGO all utilize the behavior data, we use the approach of projecting the appropriate reward functions that depend on the behavior data into a lower dimensional state space described in Section 4.1. We emphasize that BC-TRPO and DAPG cannot be extended to this setting as they require full state information for warm starting (in BC-TRPO and DAPG) and online fine tuning (in DAPG). We see from Figure 1b that LOGO is still capable of attaining good performance, although training duration is increased. We will further explore the value of being able to utilize behavior data with such incomplete state information in robot experiments in the next subsection.

## 5.2 TurtleBot Experiments

We now evaluate the performance of LOGO in a real-world using TurtleBot, a two wheeled differential drive robot (Amsters & Slaets, 2019). We train policies for two tasks, (i) Waypoint tracking and (ii) Obstacle avoidance, on Gazebo, a high fidelity 3D robotics simulator.

**Task 1: Waypoint Tracking:** The goal is to train a policy that takes the robot to an arbitrary waypoint within 1 meter of its current position in an episode of 20 seconds. The episode concludes when the robot either reaches the waypont or the episode timer expires. The state space of the agent are its relative $x, y$, coordinates and orientation $\phi$ to the waypoint. The actions are its linear and angular velocities. The agent receives a sparse reward of $+1$ if it reaches the waypoint, and $0$ otherwise. We created a sub-optimal Behavior policy by training TRPO with dense rewards on our own low fidelity kinematic model Python-based simulator with the same state and action space. While it shows reasonable waypoint tracking, the trajectories that it generates in Gazebo are inefficient, and its real-world waypoint tracking is poor (Figures 2 (d)-(f)). As expected, TRPO shows poor performance in this sparse reward setting and often does not attain the desired waypoint before the episode concludes, giving the impression of aimless circling seen in Figure 2(f). LOGO is able to effectively utilize the Behavior policy and shows excellent waypoint tracking seen in Figures 2 (d)-(f).

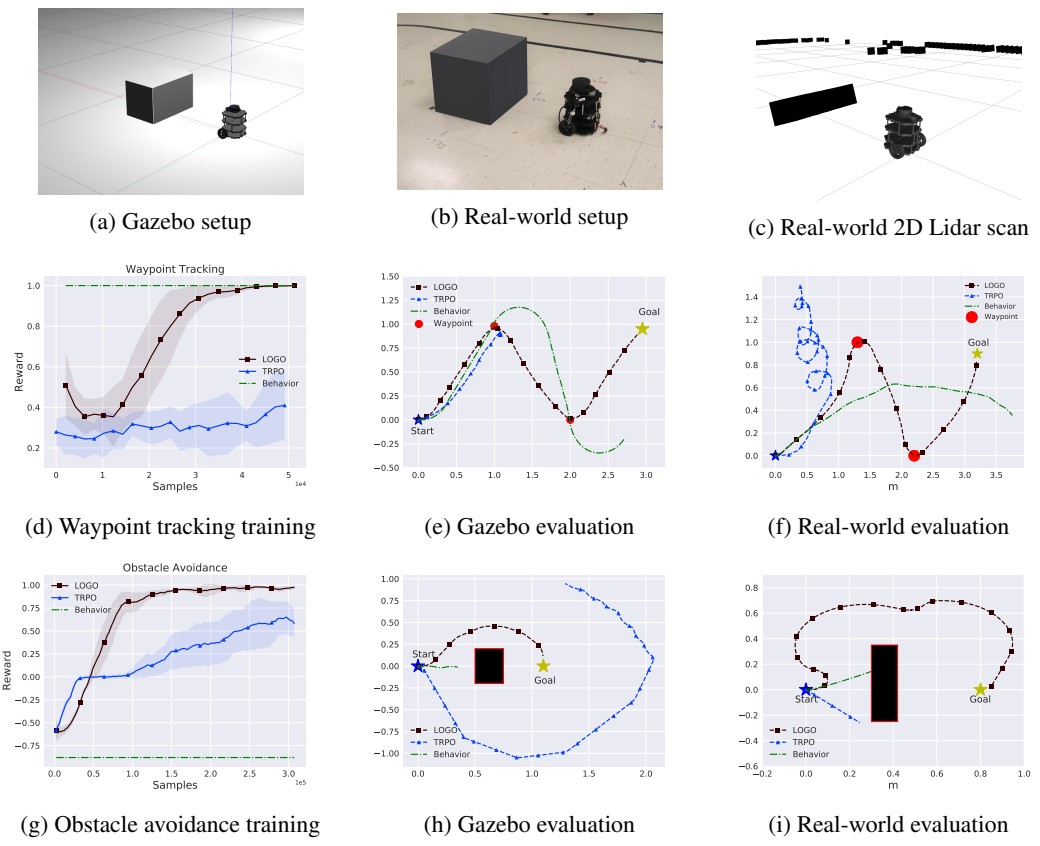

Figure 2: Training and evaluation under Gazebo simulations and real-world experiments. **(a)-(c)** show the experiment setup, with the TurtleBot added for visual reference in **(c)**. **(d)-(f)** show waypoint tracking performance. **(g)-(i)** show waypoint tracking with obstacle avoidance performance.

**Task 2: Obstacle Avoidance:** The goal and rewards are the same as in Task 1, with a penalty of $-1$ for collision with the obstacle, shown in Figure 2 (a)-(c). The complete state space is now augmented by a 2D Lidar scan in addition to coordinates and orientation described in Task 1. However, the Behavior policy is still generated via the low fidelity kinematic simulator *without* the obstacle, i.e., it is created on a lower dimensional state space. As seen in Figures 2 (g)-(i), this renders the Behavior policy impractical for Task 2, since it almost always hits the obstacle in both Gazebo and the real-world. However, it does possess information on how to track a waypoint, and when combined with the full state information, this nugget is utilized very effectively by LOGO to learn a viable policy as seen in Figures 2 (g)-(i). Further, TRPO in this sparse reward setting does poorly and often collides with the obstacle in real-world experiments as seen in Figure 2 (i).

## 6 CONCLUSION

In this paper, we studied the problem of designing RL algorithms for problems in which only sparse reward feedback is provided, but offline data collected from a sub-optimal behavior policy, possibly with incomplete state information is also available. Our key insight was that by dividing the training problem into two steps of (i) policy improvement and (ii) policy guidance using the offline data, each using the concept of trust region based policy optimization, we can both obtain principled policy improvement and desired alignment with the behavior policy. We designed an algorithm entitled LOGO around this insight and showed how it can be instantiated in two TRPO-like steps. We both proved analytically that LOGO can exploit the advantage of the behavior policy, as well as validated its performance through both extensive simulations and illustrative real-world experiments.

## 7 ACKNOWLEDGEMENT

This work was supported in part by the National Science Foundation (NSF) grants NSF-CRII-CPS-1850206, NSF-CAREER-EPCN-2045783, NSF-CPS-2038963 and NSF-CNS 1955696, and U.S. Army Research Office (ARO) grant W911NF-19-1-0367. Any opinions, findings, and conclusions or recommendations expressed in this material are those of the authors and do not necessarily reflect the views of the sponsoring agencies.

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

## A   Useful Technical Results

We will use the well known Performance Difference Lemma in our analysis (Kakade & Langford, 2002)

**Lemma 3** (Performance Difference Lemma). *For any two policies $\pi$ and $\tilde{\pi}$,*

$$J_R(\pi) - J_R(\tilde{\pi}) = (1 - \gamma)^{-1} \, \mathbb{E}_{s \sim d^\pi, a \sim \pi(s,\cdot)}[A_R^{\tilde{\pi}}(s,a)]$$

The following result gives a bound on the difference between discounted state visitation distributions of two policies in terms of their average total variation difference.

**Lemma 4** (Lemma 3, (Achiam et al., 2017)). *For any two policies $\pi$ and $\pi'$,*

$$\left\| d^\pi - d^{\pi'} \right\|_1 \le 2\gamma(1-\gamma)^{-1} \, D_{\mathrm{TV}}^\pi(\pi, \pi')$$

We will use the following well known result.

**Lemma 5.** *For any two policies $\pi$ and $\pi'$, $D_{\mathrm{TV}}^\pi(\pi, \pi') \le \sqrt{D_{\mathrm{KL}}^\pi(\pi, \pi')/2}$.*

*Proof.* From Pinsker's inequality, $D_{\mathrm{TV}}(p,q) \le \sqrt{D_{\mathrm{KL}}(p,q)/2}$ for any two distributions $p, q$. Combining this with Jensen's inequality, we get $D_{\mathrm{TV}}^\pi(\pi, \pi') \le \sqrt{D_{\mathrm{KL}}^\pi(\pi, \pi')/2}$. □

We will use the following result which is widely used in literature. We provide the proof for completeness.

**Lemma 6.** *Given any policy $\pi$ and a function $f_\pi : \mathcal{S} \times \mathcal{A} \to \mathbb{R}$,*

$$\mathbb{E}_{\tau \sim \pi}[\sum_{t=0}^{\infty} \gamma^t f_\pi(s_t, a_t)] = (1-\gamma)^{-1} \mathbb{E}_{s \sim d^\pi, a \sim \pi(s,\cdot)}[f_\pi(s,a)]$$

*Proof.*

$$\mathbb{E}_{\tau \sim \pi}[\sum_{t=0}^{\infty} \gamma^t f_\pi(s_t, a_t)] = \sum_{s,a} \sum_{t=0}^{\infty} \gamma^t \mathbb{P}(s_t = s, a_t = a | \pi) f_\pi(s,a)$$

$$= \sum_{s,a} \sum_{t=0}^{\infty} \gamma^t \mathbb{P}(s_t = s | \pi) \pi(s,a) f_\pi(s,a) = \sum_{s,a} \pi(s,a) f_\pi(s,a) \sum_{t=0}^{\infty} \gamma^t \mathbb{P}(s_t = s | \pi)$$

$$= (1-\gamma)^{-1} \sum_{s,a} d^\pi(s) \pi(s,a) f_\pi(s,a) = (1-\gamma)^{-1} \mathbb{E}_{s \sim d^\pi, a \sim \pi(s,\cdot)}[f_\pi(s,a)].$$

□

## B   Proof of the Results in Section 3

### B.1   Proof Lemma 1

*Proof.* Starting from the performance difference lemma,

$$J_R(\pi) - J_R(\pi_{k+1/2}) = (1 - \gamma)^{-1} \, \mathbb{E}_{s \sim d^\pi, a \sim \pi(s,\cdot)}[A_R^{\pi_{k+1/2}}(s,a)]$$

$$= (1 - \gamma)^{-1} \sum_s d^\pi(s) \sum_a \pi_{\mathrm{b}}(s,a) A_R^{\pi_{k+1/2}}(s,a)$$

$$+ (1 - \gamma)^{-1} \sum_s d^\pi(s) \sum_a (\pi(s,a) - \pi_{\mathrm{b}}(s,a)) A_R^{\pi_{k+1/2}}(s,a)$$

$$\overset{(a)}{\ge} (1 - \gamma)^{-1}\beta + (1 - \gamma)^{-1} \sum_s d^\pi(s) \sum_a (\pi(s,a) - \pi_{\mathrm{b}}(s,a)) A_R^{\pi_{k+1/2}}(s,a)$$

$$\overset{(b)}{\geq} (1-\gamma)^{-1}\beta - (1-\gamma)^{-1}\, \epsilon_{R,k+1/2}\, 2\, D_{\mathrm{TV}}^{\pi}(\pi, \pi_{\mathrm{b}})$$

$$\overset{(c)}{\geq} (1-\gamma)^{-1}\beta - (1-\gamma)^{-1}\, \epsilon_{R,k+1/2}\sqrt{2D_{\mathrm{KL}}^{\pi}(\pi, \pi_{\mathrm{b}})},$$

where $(a)$ follows from the premise that $\pi_{k+1/2}$ satisfies Assumption 1, $(b)$ is obtained by denoting $\epsilon_{R,k+1/2} = \max_{s,a} |A_R^{\pi_{k+1/2}}(s,a)|$, and $(c)$ follows from Lemma 5. $\qquad\square$

### B.2 PROOF THEOREM 1

We will first prove the following result.

**Lemma 7.** *For any two policies $\pi$ and $\tilde{\pi}$,*

$$|J_R(\pi) - J_R(\tilde{\pi})| \leq \frac{3R_{\max}}{(1-\gamma)^2}D_{\mathrm{TV}}^{\max}(\pi, \tilde{\pi}),$$

*where $R_{\max} = \max_{s,a}|R(s,a)|$.*

*Proof.* We have,

$$J_R(\pi) - J_R(\tilde{\pi}) = (1-\gamma)^{-1}\Big(\sum_{s,a} d^{\pi}(s)\pi(s,a)R(s,a) - \sum_{s,a} d^{\tilde{\pi}}(s)\tilde{\pi}(s,a)R(s,a)\Big)$$

$$= (1-\gamma)^{-1}\Big(\sum_{s,a} d^{\pi}(s)\pi(s,a)R(s,a) - \sum_{s,a} d^{\tilde{\pi}}(s)\pi(s,a)R(s,a)\Big)$$

$$+ (1-\gamma)^{-1}\Big(\sum_{s,a} d^{\tilde{\pi}}(s)\pi(s,a)R(s,a) - \sum_{s,a} d^{\tilde{\pi}}(s)\tilde{\pi}(s,a)R(s,a)\Big)$$

$$\leq (1-\gamma)^{-1}R_{\max}\left\|d^{\pi} - d^{\tilde{\pi}}\right\|_1 + (1-\gamma)^{-1}R_{\max}D_{\mathrm{TV}}^{\max}(\pi, \tilde{\pi})$$

$$\overset{(a)}{\leq} (1-\gamma)^{-1}R_{\max}2\gamma(1-\gamma)^{-1}D_{\mathrm{TV}}^{\max}(\pi, \tilde{\pi}) + (1-\gamma)^{-1}R_{\max}D_{\mathrm{TV}}^{\max}(\pi, \tilde{\pi})$$

$$\leq \frac{3R_{\max}}{(1-\gamma)^2}D_{\mathrm{TV}}^{\max}(\pi, \tilde{\pi}),$$

where $(a)$ follows from Lemma 4. The lower bound can also be obtained in a similar way. Combining both, we obtain the desired result. $\qquad\square$

*Proof of Theorem 1.* $(i)$ If $\pi_{k+1/2}$ satisfies Assumption 1, then (5) follows immediately by combining the results of Proposition 1 and Lemma 1.
$(ii)$ If $\pi_{k+1/2}$ does not satisfy Assumption 1, we can directly bound $J_R(\pi_{k+1}) - J_R(\pi_{k+1/2})$ without invoking Lemma 1 (and hence the implicit assumption that moving towards $\pi_{\mathrm{b}}$ improves the policy).

Use Lemma 7, we get

$$J_R(\pi_{k+1}) - J_R(\pi_{k+1/2}) \geq -\frac{3R_{\max}}{(1-\gamma)^2}D_{\mathrm{TV}}^{\max}(\pi_{k+1}, \pi_{k+1/2}) \geq -\frac{3R_{\max}}{(1-\gamma)^2}\delta_k, \qquad (11)$$

where the last inequality follows from the fact $D_{\mathrm{TV}}^{\max}(\pi_{k+1}, \pi_{k+1/2}) \leq \delta_k$ according to the policy guidance step (2).

Now, combining the result of Proposition 1 and the above inequality, we obtain (6). $\qquad\square$

## C PROOF OF THE RESULTS IN SECTION 4

### C.1 PROOF OF LEMMA 2

*Proof.* We follow the proof technique of the performance difference lemma (PDL) with some modifications to get the result. For any given initial state $s$,

$$V_{C_{\pi}}^{\pi}(s) - V_{C_{\tilde{\pi}}}^{\tilde{\pi}}(s) = \mathbb{E}_{\tau \sim \pi | s_0 = s}\Big[\sum_{t=0}^{\infty}\gamma^t C_{\pi}(s_t, a_t)\Big] - V_{C_{\tilde{\pi}}}^{\tilde{\pi}}(s)$$

$$= \mathbb{E}_{\tau \sim \pi | s_0 = s} [\sum_{t=0}^{\infty} \gamma^t (C_\pi(s_t, a_t) + V_{C_{\tilde{\pi}}}^{\tilde{\pi}}(s_t) - V_{C_{\tilde{\pi}}}^{\tilde{\pi}}(s_t))] - V_{C_{\tilde{\pi}}}^{\tilde{\pi}}(s)$$

$$\stackrel{(a)}{=} \mathbb{E}_{\tau \sim \pi | s_0 = s} [\sum_{t=0}^{\infty} \gamma^t (C_\pi(s_t, a_t) + \gamma V_{C_{\tilde{\pi}}}^{\tilde{\pi}}(s_{t+1}) - V_{C_{\tilde{\pi}}}^{\tilde{\pi}}(s_t))]$$

$$= \mathbb{E}_{\tau \sim \pi | s_0 = s} [\sum_{t=0}^{\infty} \gamma^t (C_{\tilde{\pi}}(s_t, a_t) + \gamma V_{C_{\tilde{\pi}}}^{\tilde{\pi}}(s_{t+1}) - V_{C_{\tilde{\pi}}}^{\tilde{\pi}}(s_t))]$$

$$+ \mathbb{E}_{\tau \sim \pi | s_0 = s} [\sum_{t=0}^{\infty} \gamma^t (C_\pi(s_t, a_t) - C_{\tilde{\pi}}(s_t, a_t))], \tag{12}$$

where $(a)$ is obtained by rearranging the summation and telescoping. The first summation in (12) can now be handled as in the proof of the PDL, and we omit the details. So, we get

$$\mathbb{E}_{\tau \sim \pi} [\sum_{t=0}^{\infty} \gamma^t (C_{\tilde{\pi}}(s_t, a_t) + \gamma V_{C_{\tilde{\pi}}}^{\tilde{\pi}}(s_{t+1}) - V_{C_{\tilde{\pi}}}^{\tilde{\pi}}(s_t))] = (1-\gamma)^{-1} \mathbb{E}_{s \sim d^\pi, a \sim \pi(s, \cdot)} [A_{C_{\tilde{\pi}}}^{\tilde{\pi}}(s, a)] \tag{13}$$

The second summation can be written as

$$\mathbb{E}_{\tau \sim \pi} [\sum_{t=0}^{\infty} \gamma^t (C_\pi(s_t, a_t) - C_{\tilde{\pi}}(s_t, a_t))] = \mathbb{E}_{\tau \sim \pi} [\sum_{t=0}^{\infty} \gamma^t (\log \frac{\pi(s_t, a_t)}{\pi_b(s_t, a_t)} - \log \frac{\tilde{\pi}(s_t, a_t)}{\pi_b(s_t, a_t)})]$$

$$= \mathbb{E}_{\tau \sim \pi} [\sum_{t=0}^{\infty} \gamma^t \log \frac{\pi(s_t, a_t)}{\tilde{\pi}(s_t, a_t)}] \stackrel{(b)}{=} (1-\gamma)^{-1} \sum_s d^\pi(s) \sum_a \pi(s, a) \log \frac{\pi(s_t, a_t)}{\tilde{\pi}(s_t, a_t)}$$

$$= (1-\gamma)^{-1} D_{KL}^\pi(\pi, \tilde{\pi}), \tag{14}$$

where $(b)$ is obtained by using Lemma 6.

Now, by taking expectation on the both sides of (12) w.r.t. the initial state distribution $\mu$, and then substituting (13) and (14) there, we get the desired result. $\square$

## C.2 PROOF OF PROPOSITION 2

Lemma 2 provides an approach to evaluate the infinite horizon return $J_{C_\pi}(\pi)$ of policy $\pi$ using the infinite horizon return $J_{C_{\tilde{\pi}}}(\tilde{\pi})$ and advantage $A_{C_{\tilde{\pi}}}^{\tilde{\pi}}$ of policy $\tilde{\pi}$. Unfortunately, the RHS of (7) contains two terms that requires taking expectation w.r.t. $d^\pi$, which means that empirical estimation requires samples according to $\pi$ (that are unavailable). This is a well known issue in the policy gradient literature and it is addressed by considering an approximation by replacing the expectation w.r.t. $d^\pi$ by $d^{\tilde{\pi}}$ (Kakade & Langford, 2002; Schulman et al., 2015). We follow the same approach with minor modifications in the context of our problem.

We will first prove the following lemmas.

**Lemma 8.** *For any two policies $\pi$ and $\tilde{\pi}$,*

$$J_{C_\pi}(\pi) - J_{C_{\tilde{\pi}}}(\tilde{\pi}) \leq \frac{1}{(1-\gamma)} \mathbb{E}_{s \sim d^{\tilde{\pi}}, a \sim \pi(s, \cdot)} [A_{C_{\tilde{\pi}}}^{\tilde{\pi}}(s, a)]$$

$$+ \frac{\sqrt{2} \gamma \epsilon_{\tilde{\pi}}}{(1-\gamma)^2} \sqrt{D_{KL}^{\tilde{\pi}}(\pi, \tilde{\pi})} + \frac{1}{(1-\gamma)} D_{KL}^{max}(\pi, \tilde{\pi}),$$

*where $\epsilon_{\tilde{\pi}} = \max_{s,a} |A_{C_{\tilde{\pi}}}^{\tilde{\pi}}(s, a)|$.*

*Proof.* We will separately bound the two terms on the RHS of (7) in Lemma 2. Firstly,

$$\mathbb{E}_{s \sim d^\pi, a \sim \pi(s, \cdot)} [A_{C_{\tilde{\pi}}}^{\tilde{\pi}}(s, a)] = \sum_s d^\pi(s) \sum_a \pi(s, a) A_{C_{\tilde{\pi}}}^{\tilde{\pi}}(s, a)$$

$$= \sum_s d^{\tilde{\pi}}(s) \sum_a \pi(s, a) A_{C_{\tilde{\pi}}}^{\tilde{\pi}}(s, a) + \sum_s (d^\pi(s) - d^{\tilde{\pi}}(s)) \sum_a \pi(s, a) A_{C_{\tilde{\pi}}}^{\tilde{\pi}}(s, a)$$

$$\leq \mathbb{E}_{s \sim d^{\tilde{\pi}}, a \sim \pi(s,\cdot)}[A^{\tilde{\pi}}_{C_{\tilde{\pi}}}(s,a)] + \left\| d^{\pi} - d^{\tilde{\pi}} \right\|_1 \epsilon_{\tilde{\pi}}$$

$$\overset{(a)}{\leq} \mathbb{E}_{s \sim d^{\tilde{\pi}}, a \sim \pi(s,\cdot)}[A^{\tilde{\pi}}_{C_{\tilde{\pi}}}(s,a)] + 2\gamma(1-\gamma)^{-1} \epsilon_{\tilde{\pi}} \, D^{\tilde{\pi}}_{\mathrm{TV}}(\pi, \tilde{\pi})$$

$$\overset{(b)}{\leq} \mathbb{E}_{s \sim d^{\tilde{\pi}}, a \sim \pi(s,\cdot)}[A^{\tilde{\pi}}_{C_{\tilde{\pi}}}(s,a)] + \sqrt{2}\gamma(1-\gamma)^{-1} \epsilon_{\tilde{\pi}} \sqrt{D^{\tilde{\pi}}_{\mathrm{KL}}(\pi, \tilde{\pi})}, \tag{15}$$

where $(a)$ follows from Lemma 4 and $(b)$ follows from Lemma 5.

Secondly, $D^{\pi}_{\mathrm{KL}}(\pi, \tilde{\pi}) \leq D^{\max}_{\mathrm{KL}}(\pi, \tilde{\pi})$. Using this fact and the inequality (15) in (7), we get the desired result. $\qquad \square$

We now make an interesting observation that $J_{C_{\pi}}(\pi)$ is a scaled version of the average KL divergence between $\pi$ and $\pi_{\mathrm{b}}$. We formally state this result below.

**Lemma 9.** *For any arbitrary policy $\pi$,*

$$J_{C_{\pi}}(\pi) = \mathbb{E}_{\tau \sim \pi}[\sum_{t=0}^{\infty} \gamma^t C_{\pi}(s_t, a_t)] = (1-\gamma)^{-1} D^{\pi}_{\mathrm{KL}}(\pi, \pi_{\mathrm{b}}) \tag{16}$$

*Proof.* We have,

$$J_{C_{\pi}}(\pi) = \mathbb{E}_{\tau \sim \pi}[\sum_{t=0}^{\infty} \gamma^t C_{\pi}(s_t, a_t)] \overset{(a)}{=} (1-\gamma)^{-1} \sum_{s,a} d^{\pi}(s)\pi(s,a)C_{\pi}(s,a)$$

$$= (1-\gamma)^{-1} \sum_{s} d^{\pi}(s) \sum_{a} \pi(s,a) \log \frac{\pi(s,a)}{\pi_{\mathrm{b}}(s,a)} = (1-\gamma)^{-1} D^{\pi}_{\mathrm{KL}}(\pi, \pi_{\mathrm{b}}),$$

where $(a)$ follows from Lemma 6. $\qquad \square$

*Proof of Proposition 2.* Using the result of Lemma 9 in the inequality given by Lemma 8, for any policy $\pi$ and $\tilde{\pi}$, we get

$$D^{\pi}_{\mathrm{KL}}(\pi, \pi_{\mathrm{b}}) \leq D^{\tilde{\pi}}_{\mathrm{KL}}(\tilde{\pi}, \pi_{\mathrm{b}}) + \mathbb{E}_{s \sim d^{\tilde{\pi}}, a \sim \pi(s,\cdot)}[A^{\tilde{\pi}}_{C_{\tilde{\pi}}}(s,a)] + \frac{\sqrt{2}\gamma\epsilon_{\tilde{\pi}}}{(1-\gamma)} \sqrt{D^{\tilde{\pi}}_{\mathrm{KL}}(\pi, \tilde{\pi})} + D^{\max}_{\mathrm{KL}}(\pi, \tilde{\pi}).$$

Now, replacing $\tilde{\pi}$ by $\pi_{k+1/2}$, we get

$$D^{\pi}_{\mathrm{KL}}(\pi, \pi_{\mathrm{b}}) \leq \alpha_k + \mathbb{E}_{s \sim d^{\pi_{k+1/2}}, a \sim \pi(s,\cdot)}[A^{\pi_{k+1/2}}_{C_{\pi_{k+1/2}}}(s,a)]$$

$$+ \frac{\sqrt{2}\gamma\epsilon_{\pi,k}}{(1-\gamma)} \sqrt{D^{\pi_{k+1/2}}_{\mathrm{KL}}(\pi, \pi_{k+1/2})} + D^{\max}_{\mathrm{KL}}(\pi, \pi_{k+1/2}), \tag{17}$$

where $\alpha_k = D^{\pi_{k+1/2}}_{\mathrm{KL}}(\pi_{k+1/2}, \pi_{\mathrm{b}})$, $\epsilon_{\pi,k} = \max_{s,a} |A^{\pi_{k+1/2}}_{C_{\pi_{k+1/2}}}(s,a)|$.

Now, for any $\pi$ that lies in the trust region $\{\pi : D^{\max}_{\mathrm{KL}}(\pi, \pi_{k+1/2}) \leq \delta_k\}$, we have $D^{\pi_{k+1/2}}_{\mathrm{KL}}(\pi, \pi_{k+1/2}) \leq D^{\max}_{\mathrm{KL}}(\pi, \pi_{k+1/2}) \leq \delta_k$. Using this in (17) gives the final result. $\qquad \square$

# D  DETAILS OF THE PRACTICAL ALGORITHM

As explained in Section 4, when $\pi_{\mathrm{b}}$ is known, obtaining $C_{\pi}(s,a) = \log(\pi(s,a)/\pi_{\mathrm{b}}(s,a))$ and estimating $A^{\pi}_{C_{\pi}}$ is straightforward. Hence, LOGO can perform the policy improvement and policy guidance step according to the update equations given in (10) by directly using $\pi_{\mathrm{b}}$. When the form of $\pi_{\mathrm{b}}$ is unknown and we only have access to the demonstration data $\mathcal{D}$ generated according to $\pi_{\mathrm{b}}$, we have to estimate $C_{\pi}(s,a)$ using $\mathcal{D}$ and $\mathcal{D}_{\pi}$, where $\mathcal{D}_{\pi}$ is the trajectory data generated according to $\pi$.

Instead of estimating $\log(\pi(s,a)/\pi_{\mathrm{b}}(s,a))$ directly, we make use of the one-to-one correspondence between a policy and its discounted state-action visitation distribution defined as $\rho^{\pi}(s,a) = d^{\pi}(s)\pi(s,a)$ (Syed et al., 2008). More precisely, we estimate $\log(\rho^{\pi}(s,a)/\rho^{\pi_{\mathrm{b}}}(s,a))$ using $\mathcal{D}$ and $\mathcal{D}_{\pi}$ instead of estimating $\log(\pi(s,a)/\pi_{\mathrm{b}}(s,a))$ directly. We can then use the powerful framework of

generative adversarial networks (GAN) (Goodfellow et al., 2014) to estimate this quantity. This can be achieved by training a discriminator function $B : \mathcal{S} \times \mathcal{A} \to [0, 1]$ as

$$\max_{B} \quad \mathbb{E}_{(s,a) \sim \rho^{\pi_b}}[\log B(s,a)] + \mathbb{E}_{(s,a) \sim \rho^{\pi}}[\log(1 - B(s,a))]. \tag{18}$$

We note that the GAN-based approach to estimate a distance metric between $\rho^{\pi}$ and $\rho^{\pi_b}$ is popular in the imitation learning literature (Ho & Ermon, 2016; Kang et al., 2018).

The optimal discriminator for the above problem is given by $B^*(s,a) = \frac{\rho^{\pi_b}(s,a)}{\rho^{\pi_b}(s,a) + \rho^{\pi}(s,a)}$ (Goodfellow et al., 2014, Proposition 1). Hence, given the discriminator function $B$ obtained after training to solve (18), we use $C_{\pi}(s,a) = -\log B(s,a)$ in the LOGO algorithm, which provides an approximation to the quantity of interest. As shown in Section 5, this approximation yields excellent results in practice.

We summarize the LOGO algorithm below.

---

**Algorithm 1** LOGO Algorithm

---

1: Initialization: Initial policy $\pi_0$, Demonstration data $\mathcal{D}$ or behavior policy $\pi_b$
2: **for** $k = 0, 1, 2, ..$ **do**
3:     Collect $(s, a, r, s') \sim \pi_k$ and store in $\mathcal{D}_{\pi_k}$
4:     **if** $\pi_b$ is known **then**
5:         $C_{\pi_k}(s,a) = \log(\pi_k(s,a)/\pi_b(s,a))$
6:     **else**
7:         Train a discriminator $B(s, a)$ according to (18) using $\mathcal{D}$ and $\mathcal{D}_{\pi_k}$
8:         $C_{\pi_k}(s,a) = -\log B(s,a)$
9:     **end if**
10:    Estimate $g_k$ and $F_k$ using $\mathcal{D}_{\pi_k}$
11:    Perform policy improvement step: $\theta_{k+1/2} = \theta_k + \sqrt{\frac{2\delta}{g_k^T F_k^{-1} g_k}} F_k^{-1} g_k$
12:    Decay $\delta_k$ (according to the adaptive rule described in Appendix F)
13:    Estimate $h_k$ and $L_k$ using $C_{\pi_k}$ and $\mathcal{D}_{\pi_{k+1/2}}$
14:    Perform policy guidance step: $\theta_{k+1} = \theta_{k+\frac{1}{2}} - \sqrt{\frac{2\delta_k}{h_k^T L_k^{-1} h_k}} L_k^{-1} h_k$
15: **end for**

---

## E    DETAILS OF TURTLEBOT EXPERIMENTS

We evaluate the performance of LOGO in a real-world setting using TurtleBot, a two wheeled differential drive robot (Amsters & Slaets, 2019). We consider two different simulators for training our policy. The first one is a simple kinematics-based low fidelity simulator. The second one is a sophisticated physics-based Gazebo simulator (Koenig & Howard, 2004) with an accurate model of the TurtleBot.

We use the low fidelity simulator to get a sub-optimal Behavior policy by training TRPO with dense rewards. The details are given in Section E.1. Gazebo simulator is used for implementing the LOGO algorithm in a sparse reward setting with complete and incomplete observations. The details are given in Section E.2.

### E.1    LOW FIDELITY KINEMATICS SIMULATOR

We design the low fidelity simulator using the OpenAI gym framework where the dynamics are governed by the following equations

$$x_{t+1} = x_t + v_t \cos(\theta_t)\Delta, \;\; y_{t+1} = y_t + v_t \sin(\theta_t)\Delta, \;\; \theta_{t+1} = \theta_t + \omega_t, \Delta,$$

where $x_t, y_t$ are the coordinates of the bot, $v_t$ is the linear velocity, $\omega_t$ is the angular velocity, and $\theta_t$ is its yaw calculated from the quaternion angles, all calculated at time $t$. $\Delta$ is the time discretization factor. We define the state $s_t$ to be the normalized relative position w.r.t. the waypoint,

i.e., $s_t = ((x_t - x_w)/G, (y_t - y_w)/G, \theta_t - \theta_t^w)$, where $x_w, y_w$ are the target coordinates of the waypoint, and $\theta_t^w$ is the target heading to the waypoint at time $t$. we define the action $a_t$ to be the linear and angular velocities, i.e., $a_t = [v_t, \omega_t]$. We discretize the action space into 15 actions.

We handcraft a dense reward function as follows,

$$r_t = \begin{cases} +10 & \text{if } |x_t - x_w| \le 0.05 \quad \text{and} \quad |y_t - y_w| \le 0.05, \\ -1 & \text{if } |x_t| \ge G \quad \text{or} \quad |y_t| \ge G, \\ -0.166 d_t - 0.3184|\theta_t^w - \theta_t| & \text{otherwise}, \end{cases}$$

where $d_t$ is a combination of cross track and along track error defined as follows,

$$d_t = \left((x_t - x_w)^2 + (y_t - y_w)^2\right)\sin^2(\theta_t^w - \theta_t) + |x_t - x_w| + |y_t - y_w|.$$

### E.2  HIGH FIDELITY GAZEBO SIMULATOR

Gazebo is a physics-based simulator with rendering capabilities and can be used for realistic simulation of the environment. It is configurable and can be used to model robots with multiple joint constraints, actuator physics, gravity and frictional forces and a wide range of sensors in indoor as well as outdoor settings. Gazebo facilitates close-to-real-world data collection from the robots and sensor models. Since it runs in real-time, it takes millions of simulation frames for an RL agent to learn simple tasks. Gazebo can also speed up simulation by increasing step size. This can however lead to loss of precision. For this project, we ran the training scheme on Gazebo in almost real time with a simulation step-size of $\Delta T = 0.001$.

The Gazebo simulation setup consists of the differential drive robot (TurtleBot3 Burger) model spawned in either an empty space or a custom space with obstacles at a predefined location. A default coordinate grid is setup with respect to the base-link of the robot model. A ROS (Stanford Artificial Intelligence Laboratory et al.) framework is instantiated using a custom-built OpenAI Gym environment which acts as an interface between the proposed RL algorithm and the simulation model. ROS topics are used to capture the state update information of the robot asynchronously in a callback driven mechanism. For the purpose of our experiments we use the following ROS topics,

1. `/odom` (for $x_t, y_t, \theta_t$): Odometry information based on the wheel encoder
2. `/cmd_vel` (for $v_t, \omega_t$): Linear and angular velocity commands
3. `/scan` (for obstacle avoidance): Scan values from the Lidar

The `/odom` and `/scan` topics are used to determine $s_t$ while `/cmd_vel` is used to output the required action $a_t$. The state space and action space for waypoint tracking task is similar to state and action space in section E.1. For obstacle avoidance tasks, we also include the distance values obtained form the Lidar as a part of the state.

### E.3  ROBOT PLATFORM: TURTLEBOT3

We use TurtleBot 3 (Amsters & Slaets, 2019), an open source differential drive robot equipped with a RP Lidar as our robotic platform for real-world experiments. We use ROS as the middleware to setup the communication framework. The bot transmits its state (position and Lidar information) information over a network to a intel NUC, which transmits back the corresponding action according to the policy being executed.

### E.4  TRAINING: WAYPOINT TRACKING

In navigation problems, we have a global planner that uses a high level map of the bot's surroundings for planning a trajectory using waypoints a unit-meter distance from each other, while the goal of the bot is to achieve these waypoints. To obtain a waypoint tracking scheme, we first train a behavior policy on the low fidelity simulator using TRPO to reach a waypoint approximately one unit distance (1m in the real world) away from its initial position. In each training episode, the bot is reset to the origin and a waypoint is randomly chosen as $x_w \sim \text{uniform}([-1, 1]), y_w = 1$. The episode is terminated if the robot reaches the waypoint, or if it crosses the training boundary or exceeds the maximum episode length. The behavior policy obtained after training in the low fidelity simulator

is then used in the LOGO algorithm training on Gazebo. LOGO is trained in Gazebo with sparse sparse rewards, where a reward of $+1$ is provided if the bot reaches the waypoint, and 0 otherwise. We evaluate the trained policy in Gazebo and the real world, by providing it a series of waypoints to track in order to reach its final goal.

### E.5 Training: Obstacle Avoidance

We train our bot for obstacle avoidance in Gazebo using the behavior policy described in the section above. Our goal is to use the skills of waypoint navigation from the behavior policy to guide and learn the skills of obstacle avoidance. The state space includes the Lidar scan values in addition to the state space described in section E.1. The /scan provides 360 values, each of these indicate the distance to the nearest object in a $1°$ sector. For the purpose of our experiments, we use the minimum distance in each $60°$ sector. This reduces the Lidar data to 6 values. We train our algorithm on Gazebo with a fixed obstacle for random waypoints. In each training episode, the bot is reset to the origin and a waypoint is generated similar to the previous section. The episode is terminated if same conditions in the previous section are satisfied or if a collision with the obstacle occurs. We demonstrate the performance of our algorithms both in Gazebo as well as the real-world.

## F Implementation Details

We implement all the algorithms in this paper using PyTorch (Paszke et al., 2019). For all our experiments, we have a two layered $(128 \times 128)$ fully connected neural network with $tanh$ activation functions to parameterize our policy and value functions. We use a learning rate of $3 \times 10^{-4}$, a discount factor $\gamma = 0.99$, and TRPO parameter $\delta = 0.01$. We decay the influence of the behavior policy by decaying $\delta_k$. We start with $\delta_0$, and we do not decay $\delta_k$ for the first $K_\delta$ iterations. For $k > K_\delta$, we geometrically decay $\delta_k$ as $\delta_k \leftarrow \alpha\delta_k$, whenever the average return in the current iteration is greater than the average return in the past 10 iterations. The rest of the hyperparameters for MuJoCo simulations, Gazebo simulation, and real-world experiments are given in table 1. In table 2 we provide details on the demonstration data collected using the behavior policy.

We implement LOGO based on a publicly available TRPO code base. We implement PofD, TRPO, BC-TRPO, and GAIL by modifying the same code base as used by LOGO. We run DAPG using code base and hyperparameters provided by the authors. For consistency, we run all algorithms using the same batch size.

| Environment | $\delta_0$ | | $\alpha$ | $K_\delta$ | Batch Size |
| --- | --- | --- | --- | --- | --- |
| | Complete Observation | Incomplete Observation | | | |
| Hopper-v2 | 0.01 | 0.02 | 0.95 | 50 | 20000 |
| HalfCheetah-v2 | 0.2 | 0.1 | 0.95 | 50 | 20000 |
| Walker2d-v2 | 0.03 | 0.05 | 0.95 | 50 | 20000 |
| InvertedDoublePendulum-v2 | 0.2 | 0.1 | 0.9 | 5 | 5000 |
| Waypoint tracking | 0.01 | | 0.95 | 5 | 2048 |
| Obstacle avoidance | 0.008 | | 0.9 | 5 | 2048 |

Table 1: Hyperparameters

| Environment | Offline $\mathcal{S}$ | | Online $\mathcal{S}$ | $\mathcal{A}$ | Samples | Average Episodic Reward |
| --- | --- | --- | --- | --- | --- | --- |
| | Complete Observation | Incomplete Observation | | | | |
| Hopper-v2 | $\mathbb{R}^{11}$ | $\mathbb{R}^{7}$ | $\mathbb{R}^{11}$ | $\mathbb{R}^{3}$ | 3869 | 1369.81 |
| HalfCheetah-v2 | $\mathbb{R}^{17}$ | $\mathbb{R}^{14}$ | $\mathbb{R}^{17}$ | $\mathbb{R}^{6}$ | 5000 | 2658.34 |
| Walker2d-v2 | $\mathbb{R}^{17}$ | $\mathbb{R}^{10}$ | $\mathbb{R}^{17}$ | $\mathbb{R}^{6}$ | 4584 | 2449.21 |
| InvertedDoublePendulum-v2 | $\mathbb{R}^{11}$ | $\mathbb{R}^{7}$ | $\mathbb{R}^{11}$ | $\mathbb{R}$ | 183 | 340.73 |
| Waypoint traking | $\mathbb{R}^{3}$ | | $\mathbb{R}^{3}$ | 15 | Policy | 1 |
| Obstacle avoidance | $\mathbb{R}^{3}$ | | $\mathbb{R}^{9}$ | 15 | Policy | $-0.88$ |

Table 2: Demonstration data details

# G  RELATED WORK

**Offline RL:** Recently, there have been many interesting works in the area of offline RL which use offline data to learn a policy. In particular, offline RL algorithms such as BEAR (Kumar et al., 2019), BCQ (Fujimoto et al., 2019), ABM (Siegel et al., 2020), and BRAC (Wu et al., 2019a) focus on learning a policy using *only* the offline data *without* any online learning or online fine-tuning. The key idea of these offline RL algorithms is to learn a policy that is 'close' to the behavior policy that generated the data via imposing some constraints, to overcome the problem of distribution shift. This approach, however, often results in conservative policies, and hence it is difficult to guarantee significant performance improvement over the behavior policy. Moreover, standard offline RL algorithms are not immediately amenable to online fine-tuning, as observed in Nair et al. (2020). LOGO is different from the offline RL algorithms in the following two key aspects. First, unlike the *offline* RL algorithms mentioned before, LOGO is an *online* RL algorithm which uses the offline demonstration data for guiding the online exploration during the initial phase of learning. By virtue of this clever online guidance, LOGO is able to converge to a policy that is significantly superior than the sub-optimal behavior policy that generated the demonstration data. Second, offline RL algorithms typically require *large amount* of state-action data *with the associated rewards*. LOGO only requires a small amount demonstration data since it is used only for the guiding the online exploration. Moreover, LOGO requires only the state-action observation and does not need the associated reward data. In many real-world application applications, it may be possible to get state-action demonstration data from human demonstration or using a baseline policy. However, it is difficult to assign reward values to these observation.

Advantage Weighted Actor-Critic (AWAC) algorithm (Nair et al., 2020) propose to accelerate online RL by leveraging offline data. This work is different from our approach in four crucial aspects. $(i)$ AWAC requires offline data *with associated rewards* whereas LOGO requires only the state-action observations (not the reward data). In many real-world applications, it may be possible to get state-action demonstration data from human demonstration or using a baseline policy. However, it may be difficult to assign reward values to these observations, especially in the sparse reward setting. $(ii)$ AWAC explicitly mentions that it leverages *large amounts of offline data* whereas LOGO relies on small amount of sub-optimal demonstration data. $(iii)$ LOGO gives a novel and theoretically sound approach using the double trust region structure that provides provable guarantees on its performance. AWAC algorithm does not give any such provable guarantees. $(iv)$ LOGO can be easily extended to the setting with incomplete state information where as AWAC is not immediately amenable to such extension.

There are also many recent works on addressing multiple aspects of IL and LfD, including combining IL and RL for long horizon tasks (Gupta et al., 2019), learning from imperfect (Gao et al., 2018; Jing et al., 2020; Wu et al., 2019b) and incomplete (Libardi et al., 2021; Gülçehre et al., 2020) demonstrations. Their approaches, performance guarantees and experimental settings are different from that of our problem and proposed solution approach.

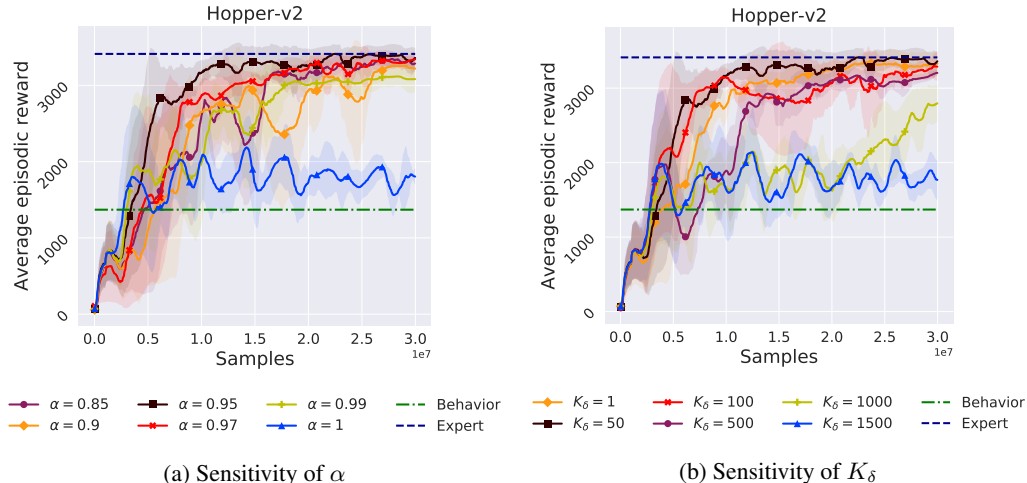

(a) Sensitivity of $\alpha$  (b) Sensitivity of $K_\delta$

# H  SENSITIVITY ANALYSIS OF $\alpha$ AND $K_\delta$

We run LOGO for different values of $\alpha$ and $K_\delta$ by keeping the other parameters fixed on the Hopper-v2 environment. We observe from Figure 3a that the performance of LOGO is not sensitive to perturbations in $\alpha$. When $\alpha = 1$, we do not decay the influence of the behavior data, this results in a performance close to the behavior data which is inline with our intuition. We observe from Figure 3b that LOGO is not sensitive to perturbations in $K_\delta$ as well. We observe that the performance of LOGO is similar for $K_\delta = 1, 50, 100$. This because we only decay $\delta_k$ whenever the average return in the current iteration is greater than the average return in the past 10 iterations. We further note that $K_\delta$ controls the iteration from which we begin decaying the influence of the behavior data, this can be clearly seen for $K_\delta = 500, 1000$ where the performance rises as soon as the decaying begins.

