# OpenReview forum: "Reinforcement Learning with Sparse Rewards using Guidance from Offline Demonstration"
_ICLR.cc/2022/Conference — ICLR 2022 Spotlight_

### Official Review · Reviewer_DXFH · 2021-10-30

**Correctness:** 4
**Technical Novelty And Significance:** 3
**Empirical Novelty And Significance:** 2
**Recommendation:** 8
**Confidence:** 3

**Main Review:**

This paper studies the problem of how to use demonstration data to better aid reinforcement learning with sparse reward, which is an important task setting that will have a wide impact on application of RL. Prior work in this direction has focused on directly imitating the demos or only using the demonstrated states to constrain exploration. The proposed LOGO algorithm leverages the per-state action distribution as a constraint for guiding policy search. The authors also show that an important advantage of LOGO is that the demonstration data need not cover the entire state space, and the algorithm is capable of matching distributions in down-projected state spaces.  This advantage allows the algorithm to be applicable to a wider range of demonstration types than pure imitation-based algorithms do. The authors performed a theoretical analysis on the algorithm and conducted experiments in a range of different tasks, including a navigation task on a real robot. The writing of this paper is clear and well organized for the readers to follow.


One weakness of the paper is that it misses one important baseline method, which is to warm-start the policy using behavior cloning with demonstration data and then run RL in the similar manner. This baseline uses the same amount of auxiliary data (i.e. the demonstrations) and potentially needs less amount of time interacting with the environment since the supervised BC step is offline. Current baseline methods do not fully leverage the data available in this problem setting. TRPO ignores the demonstrations, GAIL ignores the RL reward, and PofD only uses the state distribution instead of the state-action distribution from the demonstrations. Therefore, adding the baseline described above is important to make an apples-to-apples comparison between BC and using demos as guidance. Apparently warm-start with BC does not offer the generalizability to partially-observed demonstrations and therefore LOGO still has its unique advantages even if it is not able to outperform this baseline


**Summary Of The Paper:**

This paper presents the learning online with guidance offline (LOGO) algorithm that leverages demonstration data to constrain policy search for reinforcement learning with sparse reward such that the initial exploration phase is guided. Experiments in locomotion tasks in simulated domains and a navigation task on a real robot demonstrates LOGO is more data-efficient compared to a range of baseline methods.


**Summary Of The Review:**

This paper studies an important problem of how to leverage demonstrations for reinforcement learning and proposes a novel method that is demonstrated to outperform a set of baselines in various domains. It makes the case for how distribution matching should be used instead of pure supervised learning of demonstrations and shows an unique advantage of doing so, which is the capability of dealing with partially observable states. I recommend acceptance of this paper but would love to see another baseline added to the experiments as I mentioned in my review above.

---

> ### Author Response · Authors · 2021-11-19
> **Response to Reviewer #5( DXFH)**
>
> We thank the reviewer for the thoughtful comments and suggestions. We are also grateful for the reviewer's comment that  "I recommend acceptance of this paper but would love to see another baseline added to the experiments as I mentioned in my review''. In our revised manuscript, we have  included the baseline comparison as suggested by the reviewer. Please see our response below.
>
> **Q1**. *"One weakness of the paper is that it misses one important baseline method, which is to warm-start the policy using behavior cloning with demonstration data and then run RL''*
>
> **Response:** In our revised manuscript, we have included the comparison with the baseline method suggested by the reviewer.  In addition to this, we also have included another baseline for comparison, namely, the DAPG algorithm (Rajeswaran et al., 2018). Our finding is that LOGO outperforms these other approaches, either quite significantly in several cases, or is comparable in a few cases.  Hence, these comparisons do not materially impact our main conclusions regarding the value of LOGO.   We have now included these baseline comparisons in our paper, please see Section 5.1 and Appendix F.
>
>
> We also would like to emphasize that, as the reviewer has correctly observed, the warm-starting with behavior cloning and DAPG will not immediately generalize to the setting where the demonstration data only contains a censored version of the true state (incomplete state information). Quite different from this, LOGO can be easily extended to such settings, as we have demonstrated using MuJoCo simulations (Section 5.1) and real-world physical robot (TurtleBot) experiments.

---

### Official Review · Reviewer_FWYZ · 2021-10-31

**Correctness:** 4
**Technical Novelty And Significance:** 2
**Empirical Novelty And Significance:** 3
**Recommendation:** 6
**Confidence:** 4

**Main Review:**

The paper is well-written and clearly states the problem, motivation and goals, and the approach taken to solve the problem at hand. Formal explanations and methods are presented rigorously and are supplemented with helpful intuitive explanations following each major block of formal mathematical language. The exploration of learning from demonstrations to improve learning in challenging settings (sparse rewards, long horizons, etc) is, in my mind, an important concept that also has ~immediate practical applications and implications, and the method presented here adds to the toolbox of ideas on how this can be approached.


Major concerns / criticisms / discussion
----------------------------------------

* From a quick read of the Kang et al. 2018 is indeed appears to be very close to the idea (at least in spirit) presented in this work. Authors mention how their approach is fundamentally different in terms of methodological approach, however I think the work would benefit from a bit more detailed exploration of the differences in consequences of applying POfD vs. LOGO. If we imagine a situation where both algorithms were presented with the same data (same random seed, same everything) -- what would be crucial difference between the two policies produced by the two methods? Maybe differences in value estimation? Degree of deviation from the behavior policy? What are the arguments one could present to decide "I will choose LOGO over POfD, because ..."?

* Conceptually there does not seem to be anything that would prevent using LOGO in a large-state-space setting, for example using RGB(D) inputs from a robot. Did the authors attempt LOGO in such an environment? Is the "partial observation" trick *required* to learn in such an environment or we would see the benefits of LOGO even when working with full state directly? If you did try this then were the results positive or negative, and if you did not, then what are your thoughts on feasibility and performance expectations?


Questions
---------

* general: How hard would it be to adapt the same idea to other algorithms, that do not have the build-in notion of a trust region? For example SAC and QR-DQN? Or another way to phrase the question: how specific is the proposed method (conceptually) to TRPO. Could including some kind of a penalty based on KL divergence on top of other algorithm also work, or this would be unlikely / highly non-trivial to achieve?

* page 1 par 2: "natural candidate to begin with is the framework of policy gradient algorithms <...> performs really well in the dense reward setting" -- Unclear why this is a natural choice while in the previous paragraph we were discussing sparse rewards. Wouldn't it be more natural to first consider something that works well with sparse rewards?

* page 2 par 1: "Our choice of the TRPO " -- Why not the superior methods based on similar principles like PPO?

* page 3 assumption 1: What does beta stand for in this equation?

* page 9 "task 2": What is the dimension of the state space when lidar data is included?

Minor remarks
-------------
* page 1 par 2: "behavior data might only contain measurements of a subset of the true state" -- It was a bit unclear why that would be the case. If I understand correctly this isn't something that happens often. Was this included in the manuscript because the authors have observed that LOGO works well for situations like this? If that is the case perhaps the sentence can be reformulated to say that, without the implication that partial state is something that tends to happen often.

* sec 5.1 par 1: "moves forward over 2, 20, and 2 units from" -- at typo?

**Summary Of The Paper:**

In a regime where we have access to offline behavior data from a suboptimal policy (a heuristic, human demonstrations, etc) we can use that data and "trust region"-based methods (TRPO in this work) to nudge exploration in the right direction by keeping the learnable policy close to the behavioral one. While there are several works exploring this general idea, the particular way presented in this work is novel and achieves better results on a set of MuJoCo environment. The approach is also shown to work on a physical robot (Turtlebot).

**Summary Of The Review:**

I find the paper's goals clear and interesting and the execution faithful and technically sound. This work makes a curious step in the direction of mixing together online learning and pre-existing policy data and I think would be an interesting result for others who do research in this area. I wouldn't mind seeing this work presented at the conference.

---

> ### Author Response · Authors · 2021-11-19
> **Response to Reviewer #4 (FWYZ) (2/2)**
>
> **Q6.** *"What is the dimension of the state space when lidar data is included?''*
>
> **Response:** The resulting state dimension is 9. This is explained in Appendix E.5 and Appendix F, where we have included a detailed description about this.
>
>
> **Q7.**  *"On page 1, para 2, 'behavior data might only contain measurements of a subset of the true state'. It was a bit unclear why that would be the case''*
>
> **Response:**  The motivation behind incomplete state information is that empirical policies are often applied to systems in the field with only a limited number of sensors for measurements.  For example, it is relatively easy to obtain location and control inputs of a human driven car, but much more difficult to outfit it with Lidars and cameras to capture is true state.  Our approach is able extract useful information from such readily available but incomplete data.

---

> ### Author Response · Authors · 2021-11-19
> **Response to Reviewer #4 (FWYZ) (1/2)**
>
> We thank the reviewer for the thoughtful comments and suggestions. We are encouraged by the fact that the reviewer found our work  as "clear and interesting and the execution faithful and technically sound''. We give a detailed response to the specific comments below.
>
> **Q1.** *On the comparison with the PofD algorithm (Kang et al., 2018)*
>
> **Response:** PofD uses the idea of reward shaping using demonstration data. An intrinsic reward function is estimated from the demonstration data using a discriminator and this is used to modify the native reward function. A standard RL algorithm  is then used to learn the policy corresponding to the modified reward function. Very different from this, LOGO follows a principled approach of using the offline demonstration data for  guiding the online exploration during the initial phase of learning.   Moreover, this principled approach enables us to provide analytical performance guarantees while achieving superior performance in a number of benchmark problems. LOGO achieves  superior empirical performance in all the  four MuJoCo benchmark environments  we used, please Figure.~1 in Section 5. We emphasize that, for these experiments both algorithms were presented with the same data, used the same policy and value function structure, and were implemented using the same code base as mentioned by the reviewer. We have also demonstrated the effectiveness of LOGO in the  setting where the demonstration data only contains a censored version of the true state (incomplete state information), please the experiment results on a  real-world physical robot (TurtleBot)  described in Section 5.2.  So, one would choose LOGO over PofD solely based on its superior performance in a variety of challenging settings.  In addition to the empirical performance,  the principled double trust region structure of LOGO gives a provable performance characterization of the algorithm  in terms of the  lower bound on the performance improvement at each step.
>
>
> **Q2.**  * " $\ldots$ using LOGO in a large-state- space setting, for example using RGB(D) inputs from a robot. $\ldots$ what are your thoughts on feasibility and performance expectations''
>
> **Response**:  Conceptually, nothing prevents LOGO  from using very large dimensional input states, such as the camera images from  the mobile robots. In terms of practical implementation, using an image as the input state may require a  convolutional neural network (CNN) instead of a fully connected neural network that we used in the current implementation. Since our main focus in this work is to introduce the novel idea of 'learning online with guided exploration using offline data'  and demonstrate  its superior theoretical and empirical performance, we first performed experiments on several benchmark MuJoCo environments and did not use camera images as states. However, we note that our experiments on the real-world physical robot (TurtleBot) actually used a Lidar image. But to focus on the key idea of extension to the incomplete state information, we only considered a pre-processed version of this image, which reduced the Lidar image to a 6  dimensional state vector that indicates the presence of obstacles.
>
> **Q3.**  *"How hard would it be to adapt the same idea to other algorithms, that do not have the build-in notion of a trust region''*
>
> **Response:** LOGO algorithm has two steps, namely, the Policy Improvement  and Policy Guidance. The policy improvement step can be achieved by one step of any on-policy or off-policy  RL algorithm such as  PPO or SAC, without any real need of a trust region. The policy guidance step, however, crucially depends on the fact that $\pi_{k+1}$ remains close to $\pi_{k+\frac{1}{2}}$ where the distance is controlled by a decaying $\delta_{k}$. Moreover, this step also requires an on-policy algorithm. So, we used the natural and popular idea of trust region method here. As the reviewer suggested, it may be possible to include a penalty based on the KL divergence in the objective and use another standard RL algorithm for achieving policy guidance step. We would like to emphasize that, while the above modifications may give a reasonable empirical performance, it is likely to be very difficult to give a provable guarantee (similar to Theorem 1) on the resulting algorithm's performance. Analytical tractability is the major reason why we selected a TRPO-like approach for both steps.
>
> **Q4**.  *On "Our choice of the TRPO''*
>
> **Response:** Please see our response to the above question.
>
> **Q5.**  *"What does $\beta$ stand for in Assumption 1''*
>
> **Response:** Here $\beta$ essentially represents the positive advantage of using $\pi_{{b}}$ instead of using $\pi_{k}$. Please see the paragraph below Assumption 1, where we have explained this.

---

### Official Review · Reviewer_KhX3 · 2021-11-01

**Correctness:** 3
**Technical Novelty And Significance:** 3
**Empirical Novelty And Significance:** 2
**Recommendation:** 6
**Confidence:** 4

**Main Review:**

This paper provides a novel formulation of how to keep an RL policy close to a behavior policy. Instead of directly performing some sort of constrained optimization (maximizing the expected sum of rewards subject to the constraint being $ KL(\pi||\pi_b) < \epsilon $), the authors formulate the problem as another RL problem with the reward $ \frac{log(\pi(a|s))}{log(\pi_b(a|s))} $ while ensuring that the updated policy doesn't stray too far from the current RL policy iterate (with a KL constraint term that has a decaying coefficient). This approach is certainly novel and an intriguing way to optimize under this constraint and the empirical results certainly do appear impressive.

However, I do have several concerns.
1. This paper makes no mention of prior work that has also noted this problem of attempting to constrain the RL policy to remain close to the behavior policy via the same constraint mentioned in this work: $ KL(\pi||\pi_b) < \epsilon $. In the offline RL literature, there has been a large body of work that has attempted to optimize this constraint in different ways: (BEAR, BCQ, ABM, BRAC, MPO). I believe these, and many other works in this area should be included with a discussion of their tradeoffs in the related works section.
2. Empirically, this work is missing several clear comparisons. One: training a behavior policy using the offline data (do Behavior cloning) and simply initializing TRPO from this policy to bootstrap learning. This is a simple solution used as a baseline in prior work (see RPL) and should be included as a point of comparison. Two: DAPG. This method can in principle use suboptimal data and has been empirically validated to work on challenging sparse reward tasks (though with the use of expert demos). Three: AWAC. This method was designed explicitly for the scenario in which a large suboptimal dataset is provided (see results on D4RL) and a small amount of online finetuning is required. Comparison with this method is critical. Four: Comparison against behavior cloning with reward filtering. (simply train a behavior cloning policy with only 10%/20% of the optimal data as ranked by the return of the offline trajectories). This method has been demonstrated to outperform many recent offline RL algorithms as seen in Table 3 of the Decision Transformer paper.
3. Empirical evaluation environments. The method is mostly evaluated on relatively simple Gym tasks modified to include sparse rewards. The RL community already has much more challenging sparse reward environments in which similar methods have been tested: the Adroit suite of sparse reward hand manipulation tasks. Evaluating if the proposed method's results are relevant should entail a comparison against prior work (AWAC and DAPG at the very least) on these much more challenging robotic control tasks.
4. nitpicks:
* upperbound should be upper bound, lowerbound should be lower bound
* End of related work: “Their approaches, performance guarantees and experimental settings are different from that of our problem and proposed solution approach.” Please explain how this is the case

Citations:
BEAR: Aviral Kumar, Justin Fu, George Tucker, and Sergey
Levine. Stabilizing Off-Policy Q-Learning via Bootstrapping Error Reduction. In Neural Information Processing
Systems (NeurIPS), 2019.

BCQ: Scott Fujimoto, David Meger, and Doina Precup. OffPolicy Deep Reinforcement Learning without Exploration.
In International Conference on Machine Learning (ICML),
2019.

ABM: Noah Y. Siegel, Jost Tobias Springenberg, Felix
Berkenkamp, Abbas Abdolmaleki, Michael Neunert,
Thomas Lampe, Roland Hafner, Nicolas Heess, and
Martin Riedmiller. Keep doing what worked: Behavioral
modelling priors for offline reinforcement learning, 2020

BRAC: Yifan Wu, George Tucker, and Ofir Nachum. Behavior
Regularized Offline Reinforcement Learning. 2020.

MPO: Abbas Abdolmaleki, Jost Tobias Springenberg, Yuval
Tassa, Remi Munos, Nicolas Heess, and Martin Riedmiller. Maximum a Posteriori Policy Optimisation. In
International Conference on Learning Representations
(ICLR), pp. 1–19, 2018.

RPL: Abhishek Gupta, Vikash Kumar, Corey Lynch, Sergey
Levine, and Karol Hausman. Relay Policy Learning: Solving Long-Horizon Tasks via Imitation and Reinforcement
Learning. In Conference on Robot Learning (CoRL),
2019

Decision Transformer: Chen, Lili, Kevin Lu, Aravind Rajeswaran, Kimin Lee, Aditya Grover, Michael Laskin, Pieter Abbeel, Aravind Srinivas, and Igor Mordatch. "Decision transformer: Reinforcement learning via sequence modeling."


**Summary Of The Paper:**

This paper aims to improve the performance of RL in sparse reward settings via the guidance of sub-optimal demonstrations. The idea proposed by this work is to modify TRPO to restrict its updates to remain close to the behavior policy that generated the offline dataset, while decaying this constraint over time to enable the RL policy to improve upon the behavior policy in an online fashion. The authors do so by minimizing the KL divergence between the current RL policy and the behavior policy while not moving too far away from the current RL policy. The method is evaluated on openAI gym style tasks as well as a real robot navigation task.

**Summary Of The Review:**

In general, while the work has (at first glance) impressive empirical results as well as a novel formulation of the KL constraint between the RL policy and the behavior policy, there are quite a few concerns that would need to be addressed with direct justification and experimentation in order to recommend acceptance. Hence, I currently recommend rejection, but I am open to updating my score if all of my concerns can be addressed.

---

> ### Author Response · Authors · 2021-11-19
> **Response to Reviewer #3 (KhX3)**
>
> We thank the reviewer for the thoughtful comments and suggestions. We are also grateful for their comment that "I am open to updating my score if all of my concerns can be addressed''. The key suggestions by the reviewer are: $(i)$ include a discussion on the differences and similarities with offline RL algorithms such as BEAR, BCQ, ABM, BRAC, MPO,  $(ii)$ include empirical comparison with other baseline algorithms, and $(iii)$ include empirical evaluation environments different from the  MuJoCo environments. We are glad to report that we  have now addressed all these suggestions in our revised manuscript. Please see our detailed response below.
>
> **Q1.** *On the comparison with offline RL works such as  BEAR, BCQ, ABM, BRAC, MPO.*
>
> **Response**: Thank you for pointing this out. We have included a short comparison  in Section 1.1. of our revised paper. We have also included a detailed comparison in Appendix G of the revised manuscript.
>
> **Q2**. *On empirical comparison with  BC+TRPO,  DAPG, AWAC, and  BC+reward  filtering.*
>
> **Response:** Thank you for this suggestion. We have now included two additional baseline comparison algorithms, BC+TRPO and DAPG, in our revised manuscript according to the reviewer's suggestion. Our finding is that LOGO outperforms these other approaches, either quite significantly in several cases, or is comparable in a few cases.  Hence, these comparisons do not materially impact our main conclusions regarding the value of LOGO.   We have now included these baseline comparisons in our paper, please see Section 5.1 and Appendix F.
>
> Advantage Weighted Actor-Critic (AWAC) algorithm   (Nair et al., 2020) propose  to accelerate online RL by leveraging   offline data.  AWAC uses  offline data *with associated rewards* whereas LOGO uses only the state-action observations (not the reward data). Due to this, we believe that a fair comparison between AWAC and LOGO is not possible, since the offline element of AWAC is unable to utilize our sparse data. That is the reason we have not included an empirical comparison with the AWAC algorithm.
>
> Apart from this basic difference, we also would like to point out that LOGO is very different and has unique advantages compared to AWAC. In particular:   $(i)$ AWAC explicitly mentions that it leverages *large amounts of offline data with associated rewards*  (please see the Abstract and Introduction of  (Nair et al., 2020)), where as LOGO relies on small amount of sub-optimal demonstration data without . For example,  the data set for HalfCheetah-v2 experiment in (Nair et al., 2020)  "consists of 15 demonstrations from an expert policy and 100 suboptimal trajectories sampled from a behavioral clone of these demonstrations'' (please see Section III.A in (Nair et al., 2020)). Quite different from this, the same experiment in our paper uses only $5$ trajectories.  $(ii)$ LOGO has a novel and theoretically sound approach using the  double trust region structure that provides provable guarantees on its performance. AWAC algorithm does give any such provable guarantees. $(iii)$ LOGO can be easily extended to the setting with incomplete state information (as we have demonstrated in Section 5.1 and Section 5.2) whereas AWAC is not immediately amenable to such extension.
>
> BC+reward fitting approach is similar to offline RL algorithms, which also requires large *large amount* of offline state-action data *with the associated rewards*. In particular, reward filtering is done based on return of the offline trajectories. LOGO uses only the state-action observations not the reward information. So, we believe that a fair comparison is not possible. Moreover, this approach requires a large amount of offline data (to preform BC with  with only 10\%, 20\%, of the optimal data as ranked by the return). As we mentioned in the previous paragraph, LOGO only uses 5 trajectories of data without reward information.
>
> **Q3**  *On empirical evaluation environments*
>
> **Response:** We would like to emphasize that our experiments are not limited to  "relatively simple Gym tasks" as mentioned by the reviewer. In particular, we have demonstrated the performance of LOGO on a real-world physical robot (TurtleBot). Real-world robotics is an extremely challenging task, and is arguably much more difficult than any simulation environments. Since we have already demonstrated  the  performance of LOGO in four benchmark MuJoCo environments, we decided to perform experiments on a real-world robotic setting, instead of  another simulation environment. Please see the details on TurtleBot experiment described in Section 5.2 and Appendix E. Also, please see the **video of the real-world demonstration** included in the supplementary material.

---

> > ### Comment · Reviewer_KhX3 · 2021-11-22
> > **Response to the Authors**
> >
> > I would like to thank the authors for their response and for addressing many of my concerns.
> > With regards to the real world results, I am not entirely convinced of their difficulty as the authors claim given that a simple hand-coded policy could likely complete the task given that the waypoints themselves are provided to the agent.
> > I will raise my score to weak accept given the overall discussion and points raised, but I would highly recommend that the authors evaluate their method on much more difficult sparse reward control settings, such as the Adroit tasks, which as I mentioned in my original review are a standard task across several similar works.

---

### Official Review · Reviewer_FZbN · 2021-11-01

**Correctness:** 4
**Technical Novelty And Significance:** 3
**Empirical Novelty And Significance:** 4
**Recommendation:** 8
**Confidence:** 4

**Main Review:**

This paper contributes a useful and apparently novel algorithm for an important problem in the application of RL methods in real-world settings. While the algorithm, theory and experimental methodologies seem sound, a lack of comparison with several relevant algorithms makes it difficult to comprehensively assess the strength of the proposed method. Claims for the merit of the proposed method would also be strengthened if the paper included a sensitivity analysis with varying learning schedules for annealing away the guidance policy data. Without these elements, this paper in its current form falls slightly below the bar for inclusion at ICLR. If the authors can furnish these additional results (or provide compelling arguments why they are not necessary), then I would be willing to increase my score for this paper.

**Strengths:** The paper is very clear and well written, and the explanation of the algorithm is easy to follow. The experimental methodologies (selection of simulated experiments, measured metrics etc.) appear to be sound, and it is good to see experiments on a real-world robot which substantiates the claim that this method is useful for real-world RL. The fact that the theoretical version of the algorithm (before sampling based approximations) utilises the trust-region method means that this algorithm can be theoretically analysed, which is a nice property. The main assumption of the method (Assumption 1) is reasonable. In this reviewer’s opinion, one of the best strengths of the proposed LOGO method is that the double-TRPO structure allows this to be easily implemented using existing TRPO codebases. Implementation is often a stumbling point for RL algorithms, and this is a non-trivial strength of the method.

**Weaknesses:** The discussion of related work seems to be missing a few recent papers that might be relevant. These methods should also be represented in the simulated experiments to enable a fair comparison with related approaches. Specifically, in addition to Policy Optimization from Demonstration (POfD) [1] (which is cited and included in the simulated experiments), other relevant methods seem to be Advantage Weighted Actor-Critic (AWAC) [2], (which is cited but not included in experiments - the authors dismiss this method as heuristic based, but this is not a sufficient reason to exclude it from experiments), as well as Learning with supervision from Noisy Demonstrations (LfND) [3]. RL with GAN shaping [4] and the Cycle-of-Learning (CoL) framework [5] also appear to be tackling a similar problem to LOGO.

The inclusion of GAIL in the simulated experiment and pure TRPO in the simulated and real-world experiment helps situate the results, but neither of these methods could be expected to outperform LOGO given the sparse reward structure and the sub-optimal guidance data the present problem - if anything these methods serve as lower-bounds on performance that LOGO should out-perform. It would also be helpful for the reader to include a comparison with the naive strategy of behaviour cloning the sub-optimal policy, then continuing training with vanilla TRPO for some number of iterations.

As the authors acknowledge, there is a large body of work in combining online RL with offline demonstrations in various ways, and not every paper can compare against every prior method, however I believe POfD by itself does not allow a comprehensive assessment of the performance of the proposed approach.

My other concern is that the proposed algorithm depends critically on the learning schedule used to anneal away the offline policy guidance (which is described in appendix F), however as far as I can tell, the paper does not include a sensitivity analysis over this schedule (which potentially leaves the paper open to accusations of cherry picking results). Given that the performance of LOGO critically hinges on this schedule, I believe the paper needs to include an additional experiment showing how LOGO performs under variations of the schedule (e.g. sweeping values for $K_\delta$ as well as for $\alpha$).

**Queries and minor points:**

1. Is there anything that means LOGO is specifically useful for the sparse reward setting? Would it help or hinder in a dense reward setting?
2. Can the notion of the guidance policy ‘offer[ing] an advantage’ (Section 1) be quantified through information theoretic analyses?
3. Please ensure your charts are fully legible in greyscale - currently Figure 1 is not, and Figure 2 is a little difficult to read in greyscale.

---

[1] Kang, Bingyi, Zequn Jie, and Jiashi Feng. "Policy optimization with demonstrations." International Conference on Machine Learning. PMLR, 2018.

[2] Nair, Ashvin, et al. "Accelerating online reinforcement learning with offline datasets." arXiv preprint arXiv:2006.09359 (2020).

[3] Ning, Kun-Peng, and Sheng-Jun Huang. "Reinforcement learning with supervision from noisy demonstrations." arXiv preprint arXiv:2006.07808 (2020).

[4] Wu, Yuchen, Melissa Mozifian, and Florian Shkurti. "Shaping rewards for reinforcement learning with imperfect demonstrations using generative models." 2021 IEEE International Conference on Robotics and Automation (ICRA). IEEE, 2021.

[5] Goecks, Vinicius G., et al. "Integrating behavior cloning and reinforcement learning for improved performance in dense and sparse reward environments." arXiv preprint arXiv:1910.04281 (2019).


**Summary Of The Paper:**

This paper presents ‘LOGO’, an extension of the TRPO algorithm which enables additional learning guidance from offline, sub-optimal (possibly incomplete observation) demonstration data. By annealing away the learning contribution from the sub-optimal guidance policy data during training (with a learning schedule and corresponding hyper-parameter), LOGO utilises this data for guidance, rather than directly imitating it. Furthermore, because LOGO utilises the trust-region methodology, the authors are able to provide a theoretical analysis and lower bound on performance improvement each episode. The method shows promising performance on several MuJoCo continuous control tasks, as well as in a Gazebo TurtleBot simulation, which is also able to be transferred to a real-world robot.

**Summary Of The Review:**

The paper is strong but missing some key experiments, which preclude inclusion at ICLR in the present form. I would like to see one or more additional relevant comparison algorithms included in the simulated experiments, as well as a sensitivity analysis for the learning schedule, as described above. I have not checked the mathematical proofs thoroughly. I have not reviewed the code included in the attached supplementary material.

---

> ### Author Response · Authors · 2021-11-19
> **Response to Reviewer #2 (FZbN) (2/2)**
>
> **Q4.** *Minor point: "Is there anything that means LOGO is specifically useful for the sparse reward setting? Would it help or hinder in a dense reward setting?''*
>
> *Response:* LOGO algorithm is indeed general and it can  be used  both in a dense reward setting and a sparse reward setting. In particular, it will not hinder the learning in the dense reward setting. We, however, note that the key idea of `learning online with guidance offline' is especially valuable in the sparse reward setting because  the standard  RL algorithms fail to learn in such settings.
>
> **Q5.** *Minor point:  "Can the notion of the guidance policy ‘offer[ing] an advantage’ (Section 1) be quantified through information theoretic analyses?''*
>
> **Response:** Thank you for this great suggestion. This indeed seems an interesting  approach to improve the current theoretical results. We will explore this direction in our follow up work.
>
> **Q6.**  *Minor point: "Please ensure your charts are fully legible in greyscale - currently Figure 1 is not, and Figure 2 is a little difficult to read in greyscale''*
>
> **Response:** Thank you very much for your suggestion. We have now modified all the figures according to this suggestion.

---

> ### Author Response · Authors · 2021-11-19
> **Response to Reviewer #2 (FZbN) (1/2)**
>
> In the  main review and in the summary, the reviewer gave the following  two main suggestions for preparing the  paper revision/response: $(i)$ include additional relevant comparison algorithms, and $(ii)$ include sensitivity analysis. We are grateful for the reviewer's comment that *"If the authors can furnish these additional results (or provide compelling arguments why they are not necessary), then I would be willing to increase my score for this paper.''*  We are glad to report that we have now incorporated both these recommendations by the reviewer in our revised manuscript.  Please see our detailed response below.
>
> **Q1.**  *" $\ldots$ include a comparison with the naive strategy of behaviour cloning the sub-optimal policy, then continuing training with vanilla TRPO''*
>
> **Response:** We have included this baseline algorithm comparison  in our revised manuscript as suggested by the reviewer. In addition to this, we have also included the DAPG algorithm (Rajeswaran et al., 2018)  as another baseline algorithm for comparison. Our finding is that LOGO outperforms these other approaches, either quite significantly in several cases, or is comparable in a few cases.  Hence, these comparisons do not materially impact our main conclusions regarding the value of LOGO.
>
> **Q2.**  *"$\ldots$ the paper needs to include an additional experiment showing how LOGO performs under variations of the schedule (e.g. sweeping values for $K_{\delta}$ as well as for $\alpha$)''*
>
> **Response:**  As the reviewer suggested, we have now included this sensitivity analysis in our revised manuscript.  Our finding is that, within limits, LOGO is not really sensitive to these parameters, i.e., fine tuning is not needed and values within some ball-park range are adequate for all cases.  We have now included these additional experiments, please see Appendix H.
>
> **Q3.** *On the comparison with AWAC*
>
> **Response:** Advantage Weighted Actor-Critic (AWAC) algorithm   (Nair et al., 2020)
>  propose  to accelerate online RL by leveraging   offline data. This work is different from our approach in four crucial aspects. $(i)$ AWAC requires offline data *with associated rewards* whereas LOGO requires only the state-action observations (not the reward data). In many real-world  applications, it may be possible to get state-action demonstration data from human demonstration or using a baseline policy. However, there are typically no associated reward values with these observations, especially in the sparse reward setting.  $(ii)$ AWAC explicitly mentions that it leverages *large amounts of offline data* (please see the Abstract and Introduction of  (Nair et al., 2020)), where as LOGO relies on small amount of sub-optimal demonstration data. For example,  the data set for HalfCheetah-v2 experiment in (Nair et al., 2020)    "consists of 15 demonstrations from an expert policy and 100 suboptimal trajectories sampled from a behavioral clone of these demonstrations'' (please see Section III.A in (Nair et al., 2020)). Quite different from this, the same experiment in our paper uses only $5$ trajectories. $(iii)$ LOGO gives a novel and theoretically sound approach using the  double trust region structure that provides provable guarantees on its performance. AWAC algorithm does not give any such provable guarantees. $(iv)$ LOGO can be easily extended to the setting with incomplete state information where as AWAC is not immediately amenable to such extension.
>
>
> We also note that, due to the above mentioned differences, especially due to the points $(i)$ and $(ii)$, a fair comparison between LOGO and AWAC is not immediately possible. That is the reason we have not included an empirical comparison with the AWAC algorithm.
>
> We have now modified our description of AWAC in Section 1.1. of our revised paper.  Moreover, we have added a more detailed comparison in Appendix G.

---

> > ### Comment · Reviewer_FZbN · 2021-11-21
> > **Response to authors**
> >
> > I congratulate the authors on a comprehensive response to my (and it appears, the other reviewer's) comments. The authors have satisfied me that (i) AWAC is not an appropriate algorithm to compare like-for-like with LOGO (ii) the LOGO algorithm is indeed robust against varying hyperparameter settings, and (iii) LOGO does indeed show strong empirical performance against a wider selection of comparison methods.
> >
> > I particularly appreciated the adjustments to the paper being highlighted in blue to make checking the updated PDF easier. I have increased my score for this paper, and look forward to seeing LOGO developed further in future works.

---

### Official Review · Reviewer_DFHZ · 2021-11-02

**Correctness:** 3
**Technical Novelty And Significance:** 3
**Empirical Novelty And Significance:** 3
**Recommendation:** 8
**Confidence:** 4

**Main Review:**

The idea of using behavior policy to guide the reinforcement learning algorithm is promising given the fact that demonstrations are sparse to learn. While the sparsity can cause slow convergence, it is important to have a rapid algorithm for this kind of problems. The proofs provide lower bounds on reward gain between two consecutive updates. Theorem 1 shows that in the initial phase of learning the lower bound could be positive, where \beta is larger positive number. I have a concern about this theorem. In the initial phase, the KL divergence of \pi_k and \pi_b is also larger given the fact that \delta_k is decreasing. In this case, the result in Theorem 1 is not always positive which may not indicate faster learning in the initial phase.

In the experimental section, the proposed LOGO algorithm seems to have the fastest convergence compared to others. Could you give more description about convergence rates comparison from a theoretical point of view?


**Summary Of The Paper:**

This paper considers the problem of reinforcement learning with sparse reward functions obtained from offline demonstration. The authors propose a trust region policy optimization based algorithm with offline demonstration data for guidance. The proposed LOGO algorithm is proved to be efficient by a theoretical analysis showing the lower bound on the performance improvement. On benchmark datasets, the proposed algorithm performs better than the state-of-the-art approaches. An illustration is shown by implementing LOGO on a mobile robot for trajectory tracking and obstacle avoidance.

**Summary Of The Review:**

This paper proposed a novel algorithm to solve reinforcement learning problems with offline demonstration policy. The idea is interesting and nature for many real problems, especially for learning from demonstration problems. A very straightforward update rule is provided by using an estimation of KL divergence. This improves the computational efficiency of the proposed algorithm, which is very promising in real problems.  The Mujoco and TurtleBot experiments verify the performance of the algorithm with both simulation and real experiment. Overall, this paper is well written and the idea is novel and promising.

---

> ### Author Response · Authors · 2021-11-19
> **Response to Reviewer #1 (DFHZ)**
>
> We thank the reviewer for the insightful comments and questions. We are encouraged to know that the reviewer found  "the paper is well written and the idea is novel and promising'', and that they appreciated the technical analysis of our work. Below, we give a detailed response to the comments by the reviewer.
>
> **Q1**.  *"In the initial phase, the KL divergence of $\pi_k$ and $\pi_b$ is also larger given the fact that $\delta_k$ is decreasing.''*
>
> **Response**:  Theorem 1 characterizes the  performance  guarantee for the LOGO algorithm by giving a lower bound on $ J_R(\pi_{k+1}) -   J_R(\pi_{k})$  under two scenarios. When $ \pi_{k+\frac{1}{2}} $ does not satisfy Assumption 1, the lower bound is given  in Eq. (6). Please note that this is similar to the performance guarantee of the standard TRPO algorithm (Proposition 1, (Achiam et al., 2017)). In particular, the lower bound is not positive, similar to that of the TRPO performance guarantee.  When $\pi_{k+\frac{1}{2}}$ does not satisfy Assumption 1, the lower bound is given  in Eq. (5). Thus, the lower bound has the additional terms which can can add a non-negative value to the standard lower bound obtained by the TRPO approach. We agree with the reviewer's observation  that the contribution by these additional terms may not be positive always, especially if $D^\{\pi\}_{KL}  (\pi_\{k+1\},\pi_b)$ is very large and $\beta$ is very small. However, the Policy Guidance step of our algorithm, given in Eq. 2, is addressing precisely this issue, by guiding the current policy in the direction of $\pi_b$  and thus reducing  $D_\{KL\}(\pi_\{k+1\},\pi_b)$. This intuition is validated by our experiments also, which shows that LOGO is learning fast in the initial phase compared to the standard TRPO algorithm.
>
> **Q2**.  *"In the experimental section, the proposed LOGO algorithm seems to have the fastest convergence compared to others. Could you give more description about convergence rates comparison from a theoretical point of view?''*
>
> **Response**:   Thank you for noticing the fast convergence of LOGO in the initial phase compared to other algorithms and pointing out this interesting theoretical question. In general, it is very hard to a give precise rate of convergence for deep RL algorithms which use function approximation. This is why we decided to focus on providing a monotone improvement guarantee, as given in Theorem 1. Recently, there has been some interesting work on characterizing the rate of convergence of policy gradient algorithms with softmax paramaterization [1, 2]. In our future work, we will try to extend the technical approaches used in these papers for analyzing LOGO with softmax  paramaterization.
>
> [1] A. Agarwal, S. M. Kakade, J. D. Lee, and G. Mahajan, "On the theory of policy gradient methods: Optimality, approximation, and distribution shift'', *Journal of Machine Learning Research*, 22(98):1– 76, 2021.
>
> [2] J. Mei, C. Xiao, C. Szepesvari, and D. Schuurmans, "On the global convergence rates of softmax policy gradient methods'', *International Conference on Machine Learning (ICML)*, 2020.

---

### Author Response · Authors · 2021-11-19
**General Response**

We thank all the reviewers for their thoughtful comments and suggestions. We apologize for the delay in submitting our response, as we needed time to conduct all the additional experiments suggested by the reviewers. The main elements of our response are as follows:

**1. Additional experiments with relevant comparison algorithms:** Reviewers \#2, \#3, \#5 have asked for comparison with additional relevant algorithms.  We have now conducted a comparison with (i) BC-TRPO, where we warm start TRPO by performing behavior cloning (BC) on the sub-optimal behavior data, and (ii) the DAPG algorithm (Rajeswaran et al., 2018).  Our finding is that LOGO outperforms these other approaches, either quite significantly in several cases, or is comparable in a few cases.  Hence, these comparisons do not materially impact our main conclusions regarding the value of LOGO.   We have now included these baseline comparisons in our paper, please see Section 5.1 and Appendix F.

**2. Additional experiments for sensitivity analysis:** Reviewer \#2 has asked us to include additional experiments showing the sensitivity analysis for the learning schedule by changing the values for $K_{\delta}$ (the number of iterations for which the radius of the trust region in the guidance step is held constant) and $\alpha$ (the decay rate that reduces guidance by the baseline policy). Our finding is that, within limits, LOGO is not really sensitive to these parameters, i.e., fine tuning is not needed and values within some ball-park range are adequate for all cases.  We have now included these additional experiments, please see Appendix H.

**3. More discussion on related works:** Reviewer \#3 has asked us to include a comparison on offline RL algorithms. We have now included a detailed discussion on this in Appendix G. We have also included a shortened  version (due to page limits) in Section 1.1. Reviewers \#2 and \#3 have asked for a comparison with the AWAC algorithm (Nair et al., 2020).   We cannot actually compare the empirical performance of AWAC with LOGO, since (i) AWAC requires behavior data that contains reward information, whereas ours only has state-action, and (ii) AWAC explicitly requests a large behavior dataset, whereas we work with only 5 episodes of behavior data.  So the offline part of AWAC essentially does not work with our sparse dataset.  We have included a detailed comparison in the individual replies, and have also included it Appendix G. A shortened version (due to page limit) is also included in Section 1.1.

Please note that our motivation for focusing on such a sparse offline dataset is that many empirically used policies do not have any measurable reward, and we can only collect limited trajectory information. For example, we can collect location, heading and joystick inputs of a human driving a robot to a given way point, but we do not know the motivation behind the driver's actions.

Finally, as a couple of reviewers have remarked, the value of LOGO is also that it can leverage incomplete state information.  For example, in the TurtleBot experiments, the baseline robot policy does not have a Lidar or camera sensor measurements, and it does not have any obstacle information.  But LOGO is still  able to leverage this incomplete-state policy and quickly generate effective policies when Lidar inputs are included.  We would also like to point the reviewers to the **video on robot control** that we had originally included in the supplementary material, which shows how LOGO produces an efficient trajectory for attaining way points.

We have provided detailed response to each reviewer's comments. We have also modified our paper to incorporate the comments and suggestions by the reviewers. The modified parts of the paper are marked in blue.

---

### Decision · Program_Chairs · 2022-01-20

**Decision:**

Accept (Spotlight)

**Comment:**

The authors introduce a method for improving reinforcement learning in sparse reward settings. In particular, they propose to take advantage of a suboptimal behavior policy as a guidance policy that is incorporated in a TRPO-like update. The reviewers agree that this is a novel and interesting idea and given the authors' rebuttal with additional experiments, clarifications and discussions, they agreed to accept the paper. However, they also point out several flaws (e.g. evaluation on a more challenging sparse-reward task such as Adroid) that I encourage the authors to address in the final version of the paper.